# Neural oscillations in the fronto-striatal network predict vocal output in bats

**Kristin Weineck**[1,2]*, **Francisco García-Rosales**[1], **Julio C. Hechavarría**[1]*

**1** Auditory Computations lab, Institute for Cell Biology and Neuroscience, Goethe University, Frankfurt am Main, Germany, **2** Research Group Neural and Environmental Rhythms, MPI for Empirical Aesthetics, Frankfurt, Germany

* k.weineck@hotmail.de (KW); hechavarria@bio.uni-frankfurt.de (JCH)

## Abstract

The ability to vocalize is ubiquitous in vertebrates, but neural networks underlying vocal control remain poorly understood. Here, we performed simultaneous neuronal recordings in the frontal cortex and dorsal striatum (caudate nucleus, CN) during the production of echolocation pulses and communication calls in bats. This approach allowed us to assess the general aspects underlying vocal production in mammals and the unique evolutionary adaptations of bat echolocation. Our data indicate that before vocalization, a distinctive change in high-gamma and beta oscillations (50–80 Hz and 12–30 Hz, respectively) takes place in the bat frontal cortex and dorsal striatum. Such precise fine-tuning of neural oscillations could allow animals to selectively activate motor programs required for the production of either echolocation or communication vocalizations. Moreover, the functional coupling between frontal and striatal areas, occurring in the theta oscillatory band (4–8 Hz), differs markedly at the millisecond level, depending on whether the animals are in a navigational mode (that is, emitting echolocation pulses) or in a social communication mode (emitting communication calls). Overall, this study indicates that fronto-striatal oscillations could provide a neural correlate for vocal control in bats.

## Introduction

Vocalization-based interactions between broadcaster and receiver play an important role in everyday life scenarios and are highly conserved throughout the animal kingdom [1,2]. Yet, the neural circuits involved in vocal production have not been clearly delineated. Cortico-striatal networks have been identified as candidate circuits determining vocal output in mammals. Rhythmic neural activity (also known as oscillations) in striatal structures such as the caudate, putamen, and nucleus accumbens has been linked to speech production in healthy humans, to disorders such as stuttering [3], and to diseases that involve speech impairments such as Parkinson disease and Tourette syndrome [4–6]. Yet, to date, it remains largely discussed how (and if) oscillations in neural networks involving the striatum participate in the precise control of vocal motor outputs in humans and other vertebrate species.

In this article, we studied neural activity in the dorsal striatum (caudate nucleus, CN) and frontal cortex during vocalization in bats. We chose to study the fronto-striatal circuit because

**Data Availability Statement:** Data used in the manuscript to correlate brain signals to vocal production in bats are freely accessible online from the g-node database (https://doi.org/10.12751/g-node.6a0d94).

**Funding:** The project was conducted with funds provided by the German Research Foundation (DFG) to JCH (grant #275755787). The funders had no role in study design, data collection and analysis, decision to publish, or preparation of the manuscript.

**Competing interests:** The authors have declared that no competing interests exist.

**Abbreviations:** CN, caudate nucleus; dVS, difference in vector strength; FAF, frontal auditory field; IQR, interquartile range; LF, low-frequency; LFP, local field potential; LHF, low- and high-frequency; PIN, pyramidal-interneuron; PING, pyramidal-interneuron gamma; PSTH, peri-stimulus time histogram; SPL, sound pressure level; SVM, support vector machine; VS, vector strength.

there is strong evidence suggesting a role of this network in vocal production across vertebrate species. Fronto-striatal networks connect different parts of the frontal lobe with various regions of the striatum, which constitute a major input structure into the basal ganglia [7,8]. Using tractographic methods, a direct connection between the CN and the prefrontal cortex has been identified in humans and other mammals [9–12]. Both brain regions are highly connected to brain areas of the canonical vocal motor pathway. For example, in primates, the CN receives inputs from the laryngeal motor cortex [13,14]. It is also known that the frontal cortex is connected to structures participating in vocal control, such as the periaqueductal gray in the brainstem [15]. Moreover, studies examining the function of frontal and striatal regions have identified their putative role in vocalization in humans [16–18] and bats [19,20]. Likewise, in songbirds, Area X (the bird striatum) appears to be involved in vocal learning and in modulating song production in adult animals [21]. Together these studies support the involvement of frontal and striatal areas in mediating and predicting vocal output across vertebrate species.

To assess the neural dynamics during vocalization, we recorded local field potentials (LFPs) and spiking activity during the production of echolocation pulses and communication calls in bats. LFPs reflect the sum of synaptic activity in neuronal populations and slow spike components, and they represent a correlate of signals obtained with noninvasive techniques such as electroencephalography [22]. We focused on investigating synchronized neural oscillations occurring in the LFPs before and after vocal production. Oscillations are thought to enable communication between neural populations and, at least in humans and birds, they are known to be related to vocal production [23–25].

Neural oscillations can be split into different frequency bands comprising delta (1–3 Hz), theta (4–8 Hz), alpha (8–12 Hz), beta (12–30 Hz), and gamma (30–80 Hz). Empirical evidence indicates that LFP oscillations with different frequencies correlate well with distinct neural computations, motor control, and cognitive states [26]. In particular, low frequencies such as theta and alpha are known to modulate sensory processing, action selection and neuronal excitability; are implicated in cognitive control; and are involved in long-range synchrony facilitating, e.g., top-down processing [27–32]. Beta band oscillations potentially hold functions in perception, memory, and sensory processing [33–35]; are linked to motor actions in the motor cortex and striatum [36,37]; and are dysregulated in disorders such as Parkinson disease [6,38]. Gamma rhythms can be linked to selective attention, (local) neural computation, and motor control [25,27,39] and are correlated with vocalization production [24,40]. In humans, distinct oscillatory patterns and coherence across frequency bands have been found during speech production and singing [41,42].

We studied neural oscillatory activity during vocal production in bats of the species *Carollia perspicillata*. This bat species belongs to the suborder Microchiroptera, which are characterized by laryngeal echolocation, similar to human laryngeal-based speech production [43]. As bats heavily depend on their ability to vocalize in order to communicate and orient in the environment, they serve as a good animal model for studying the neural underpinnings of hearing and vocal production. Bat calls can be broadly split into two types of outputs, including echolocation pulses and communication calls such as distress and social calls (here classified as echolocation pulses versus communication calls) [44,45]. At the level of the brainstem, it is has been demonstrated that the two types of vocal outputs are distinctly controlled [46,47]. However, whether differences exist in the neural activity patterns leading to the production of both types of vocal outputs on a cortical/cortico-striatal level in bats is unknown. So far, only a very limited number of experiments were able to obtain electrophysiological recordings from vocalizing bats [48–51], even though the brain of these animals has been studied for over 50 years.

In bats, an elevated *c-fos* immunoreactivity was found in the CN of the striatum when comparing vocally active with silent animals [19]. However, the striatal neural activity patterns

related to vocal production remain unknown. The bat frontal lobe is also a rather unexplored region. Most previous experiments in bat frontal areas evaluated the auditory responsiveness of the frontal cortex and defined the frontal auditory field (FAF) [52–54]. It remains controversial whether the FAF is an analogue to the prefrontal areas found in other mammals based on morphology and connectivity [55,56]. This work will refer to the FAF when discussing the recordings from the bats' frontal lobe.

We hypothesized that the production of echolocation pulses and communication calls in bats could involve different fronto-striatal oscillatory dynamics. Echolocation and communication sounds have different purposes: the former are used to create an acoustic image of the environment (which depends on listening to echoes of the calls emitted), while the latter are uttered to convey information to other individuals. As neural oscillations have been demonstrated to be involved in a multitude of tasks (see above) and are altered in movement/speech disorders, we thought they could provide a neural correlate of vocal production. Our results show that fronto-striatal oscillations can be used to predict vocal output in bats. Vocal production correlates well with distinct inter-areal coupling in the theta band and specialized intra-areal processing mechanisms in the gamma and beta bands of LFPs. Taken together, our results present correlative evidence for the involvement of fronto-striatal circuits in motor action-pattern selection to produce different vocal outputs.

## Results

To assess fronto-striatal network activity during vocalization, 47 extracellular, paired recordings were acquired from the FAF and the CN of the dorsal striatum of four male bats. Striatal recordings were performed with linear tetrodes (electrode spacing: 200 μm), while FAF activity was measured with linear 16-channel probes (electrode spacing: 50 μm). The placement of chronically implanted tetrodes in the CN was confirmed histologically for each animal (see example Nissl section in S1 Fig). The laminar probe used for FAF measurements was introduced on each recording day. Throughout the manuscript, we will refer to different frequency bands of the LFP as theta (4–8 Hz), alpha (8–12 Hz), low beta (12–20 Hz), high beta (20–30 Hz), low gamma (30–50 Hz), and high gamma (50–80 Hz).

### Properties of bat vocalizations

Individual bats were placed in an acoustically and electrically isolated chamber and allowed to vocalize spontaneously while neural activity in the CN and FAF were simultaneously measured. A total of 39,014 spontaneously emitted calls were recorded from implanted, head-fixed animals. Most of the vocalizations recorded occurred as trains of syllables produced at short intervals (Fig 1A and 1B). Across recordings, the median calling interval amounted to 12 ± 54 ms (± interquartile range, IQR).

For analyzing neural activity related to vocalization, we focused on utterances surrounded by at least 500 ms pre- and post-time without sounds. A pool of 628 communication calls and 493 echolocation pulses remained after vocalization selection (communication: 628/16,204 [3.9%]; and echolocation: 493/22,810 [2.2%]). The main criterion used for classifying sounds into echolocation and communication was based on their spectro-temporal structure. It is known that *C. perspicillata's* echolocation pulses are short (<2 ms) downward frequency modulated and peak at high frequencies >50 kHz (see example spectrogram in Fig 1C and [57]), while communication calls cover a wider range of sound durations and contain most energy at lower frequencies, generally below 50 kHz (see examples in Fig 1D and 1E and [58,59]).

In our dataset, at the population level, call duration of both types of isolated vocalizations did not differ statistically ($p = 0.56$, Wilcoxon rank-sum test; the test considered only the

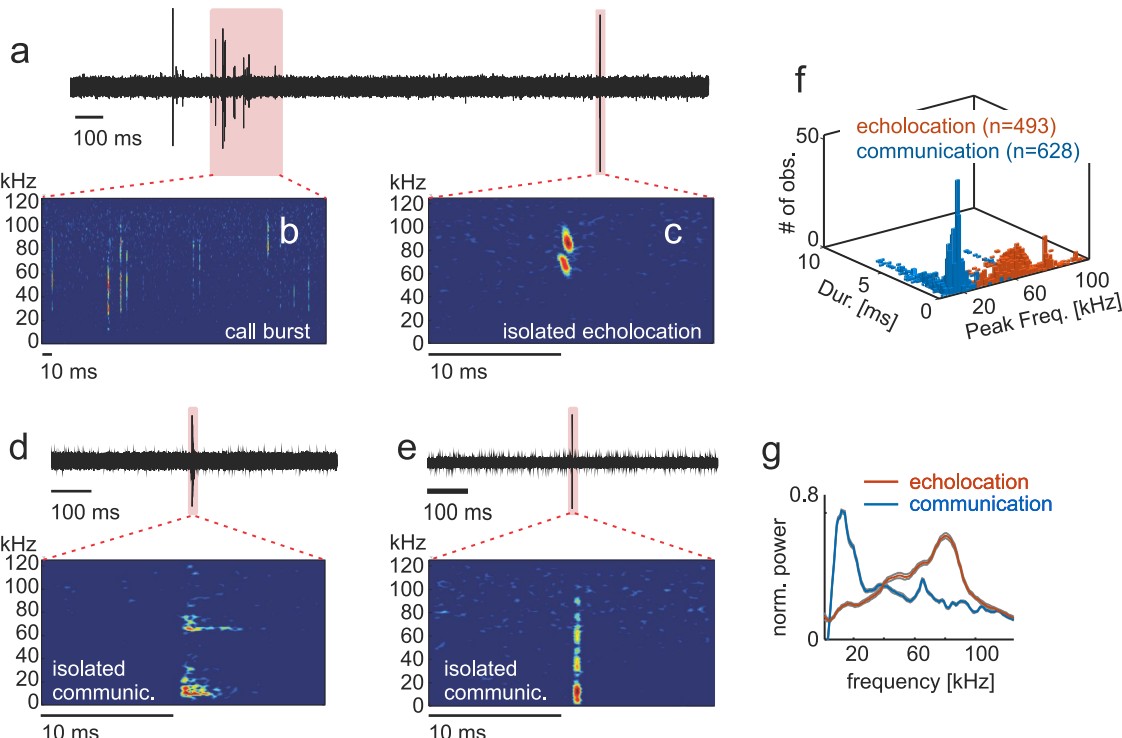

**Fig 1. Properties of echolocation pulses and communication calls produced by bats. (a)** Exemplary acoustic recording including an isolated call and a vocalization train. Zoomed-in views have been included in (**b**) and (**c**) to show spectrograms of the syllable train and the isolated echolocation pulse. Panels (**d**) and (**e**) display two further examples of isolated vocalizations (communication calls in this case). Panel (**f**) shows a combined histogram of sound duration and peak frequency in echolocation and communication sounds. Note that these two call types are well segregated in the frequency domain. The latter is also noticeable in the average call spectra shown in (**g**). Data underlying this figure can be found at https://doi.org/10.12751/g-node.6a0d94.

temporally isolated calls used for further analysis) with a median around 0.33 ms in both cases and IQR values of ±1.73 ms for communication and ±1.23 ms for echolocation (Fig 1F). As expected, peak frequency differed significantly between the two call types (echolocation, 72 kHz ± 30 kHz, and communication, 14.0 kHz ± 6 kHz, rank-sum test $p < 0.0001$, Fig 1F). In the echolocation category, differences in peak frequency across calls could be due to interindividual variability in sonar pulse design and to the use of different echolocation harmonics even by the same bat. Differences between echolocation and communication calls reported here were also evident in median spectra calculated considering all calls from each call type (Fig 1G). As frequency was used as the main distinctive feature for characterizing the two call classes, the results described in the preceding text constitute a proof-of-principle.

Communication calls were further subdivided into those that contained pronounced power only at low frequencies (<50 kHz, "LF" communication calls, $n = 319$) and those that contained pronounced energy at both low and high frequencies ("LHF" communication calls, $n = 309$, see call examples and median spectra of both communication call groups in S2 Fig). This classification of communication calls considers only the spectral structure of the sounds but does not provide information about the function of the calls uttered. The communication call category considered in this manuscript covers a broad range of vocalizations, and it might include different sound types.

Besides neural recordings during spontaneous vocalization, bats were presented with pure tones (10–90 kHz in steps of 5 kHz at a 60-dB sound pressure level [SPL] with a 10-ms duration)

to evaluate auditory responsiveness in the neural populations recorded (see frequency tuning results in S3 Fig). The acquired LFPs in both brain regions showed pronounced responses to sounds (see population-evoked responses in S3A and S3B Fig and S3I–S3K Fig), revealing a preference towards low frequencies around 15–20 kHz (best frequency distributions for both structures studied are shown in S3C and S3D Fig). Note that a preference towards low-frequency sounds does not necessarily imply a lack of responses to natural high-frequency sounds such as echolocation calls (see below). Within columns of the FAF, channels located at depths below 400 μm showed the highest auditory responsiveness, and neighboring channels had similar frequency tuning properties (see comparison of frequency tuning curves across cortical layers in S3E Fig).

## Correlating LFP oscillations with vocal output

LFPs occurring 500 ms before and after call onset were analyzed to gain insights into the involvement of fronto-striatal regions in vocalization. LFPs were filtered (1–90 Hz), demeaned, and z-normalized (see Methods). Average LFPs obtained in the CN and FAF are shown in Fig 2 (CN: Fig 2A and 2B; FAF: colormaps in Fig 2C and 2D; see also S4 Fig for recordings in one example column). Deflections in the LFP signals following the production of echolocation pulses and communication calls were evident in both brain areas. These deflections could reflect evoked responses related to the processing of the vocalizations. In the FAF, vocalization-evoked responses were strongest in deep layers (i.e., channels located at depths >400 μm)

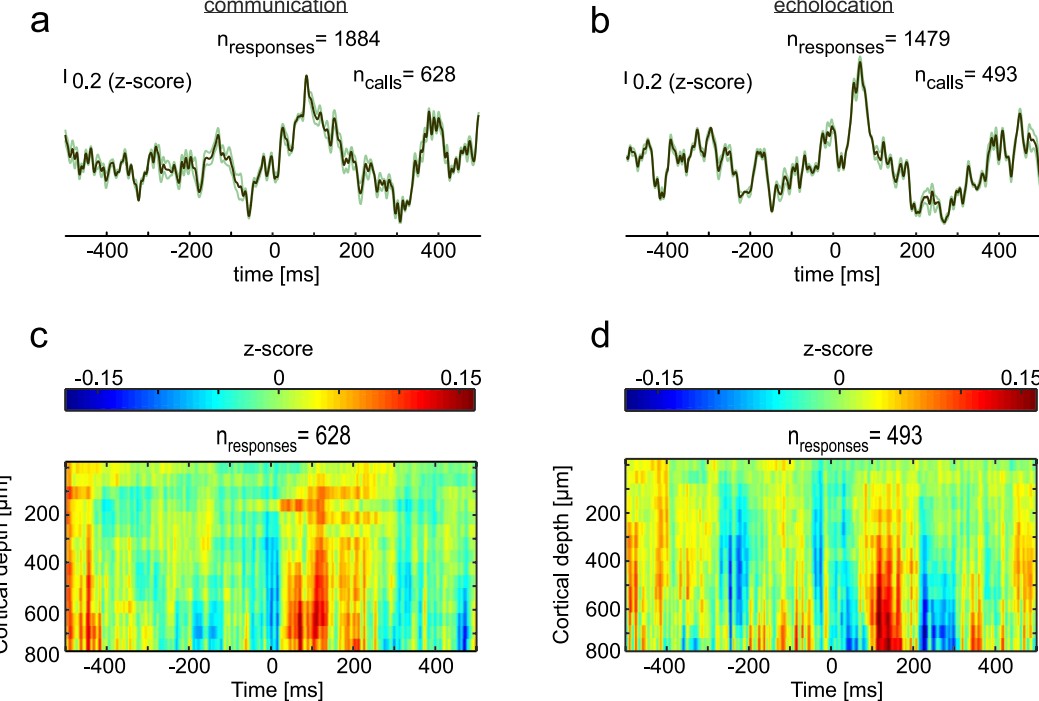

**Fig 2. LFPs during vocalization in the CN and FAF.** (**a**) Mean LFP (± SEM) of all isolated communication calls ($n = 628$) studied. Signals from all three channels of the striatum were pooled together thus rendering a higher number of responses for the striatum than for the FAF. (**b**) Mean LFP (± SEM) obtained during the production of isolated echolocation pulses (n = 493) in the striatum. (**c**) and (**d**) Colormaps showing the mean of z-scored LFPs in the FAF across cortical depths, 500 ms before and after communication calls (c) and echolocation pulses (d). Data underlying this figure can be found at https://doi.org/10.12751/g-node.6a0d94. CN, caudate nucleus; FAF, frontal auditory field; LFP, local field potential.

matching the areas of highest responsivity to pure tones (compare colormaps in Fig 2C and 2D with the colormap shown in S3 Fig; see also the example column in S4 Fig).

Next, we performed spectral analysis of the LFP signals. LFP spectrograms were calculated from bootstrapped signals based on 10,000 randomization trials for each vocalization type (see Methods). This approach allowed us to assess spectral components that are consistently time locked across vocalization trials. The striatal spectrograms followed the typical power rule by which high power occurred in the low LFP frequencies and power decreased as LFP frequency increased (Fig 3A and 3B).

When comparing both conditions (echolocation versus communication) with each other, time- and frequency-dependent variations were detected. These differences became obvious when comparing both power spectrograms using the Cliff's Delta (*d*) metric (Fig 3C). Briefly, the *d*-metric describes the effect size of group comparisons and ranges from −1 to 1, with identical groups rendering values around zero [60]. This measure was designed for nonparametric tests (in contrast to Cohen's *d*), and it quantifies how often values in one distribution are larger than values in a second distribution.

Cliff's Delta matrices revealed higher power in the gamma range of the LFP (especially frequencies >70 Hz) before communication call production in relation to the time before emission of echolocation pulses (blue areas in Fig 3C). Differences in the gamma range prior to vocalization had a medium size effect (gray contour lines in Fig 3C) following values proposed in previous studies [60]. In contrast, power in the beta range (12–30 Hz) was found to be more pronounced before and during echolocation than during communication emission. Both effects observed suggest that the power in distinct striatal LFP frequencies can be correlated with the production of different types of vocalization (beta is higher for echolocation, and gamma for communication). To portray the power of individual examples, representative single trials of LFP signals in the frequency ranges displaying the highest vocalization-dependent differences are shown in S5A–S5D Fig (CN) and S5E–S5L Fig (FAF).

Similar to the CN, spectrograms of FAF neural signals related to communication calls (Fig 4A–4D) and echolocation pulses (Fig 4E–4H) followed a power rule. Large differences could be detected when comparing the neural spectrograms obtained during echolocation and communication (Fig 4I–4L). The largest differences were found in the low- and high-gamma range, with the power being higher before and during echolocation pulses than during communication calls, especially at FAF depths below 200 μm. The latter is illustrated in Fig 4I–4L for four exemplary recording channels located at different depths and in Fig 4O for all FAF

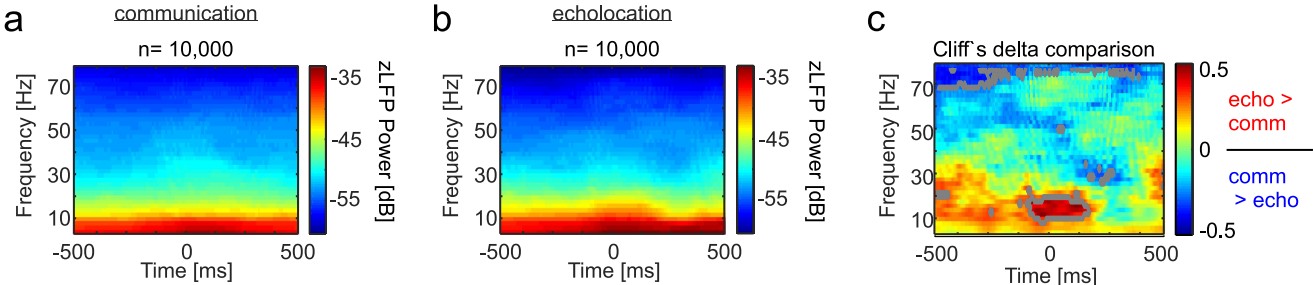

**Fig 3. Spectral differences in neural activity obtained in the CN during echolocation and communication production. (a)–(b)** Power spectrogram in the CN during communication (a) and echolocation (b). Mean values of 10,000 randomization trials are displayed in each case. **(c)** Colormap representing the Cliff's Delta values of echolocation versus communication comparisons at each time point and frequency. Gray outlined regions mark areas with a medium effect size (Cliff's Delta > 0.33 [60]). Red colors indicate more power in the LFPs during echolocation than communication. Blue regions indicate the opposite trend. The LFPs underlying this figure can be found at https://doi.org/10.12751/g-node.6a0d94. CN, caudate nucleus; LFP, local field potential.

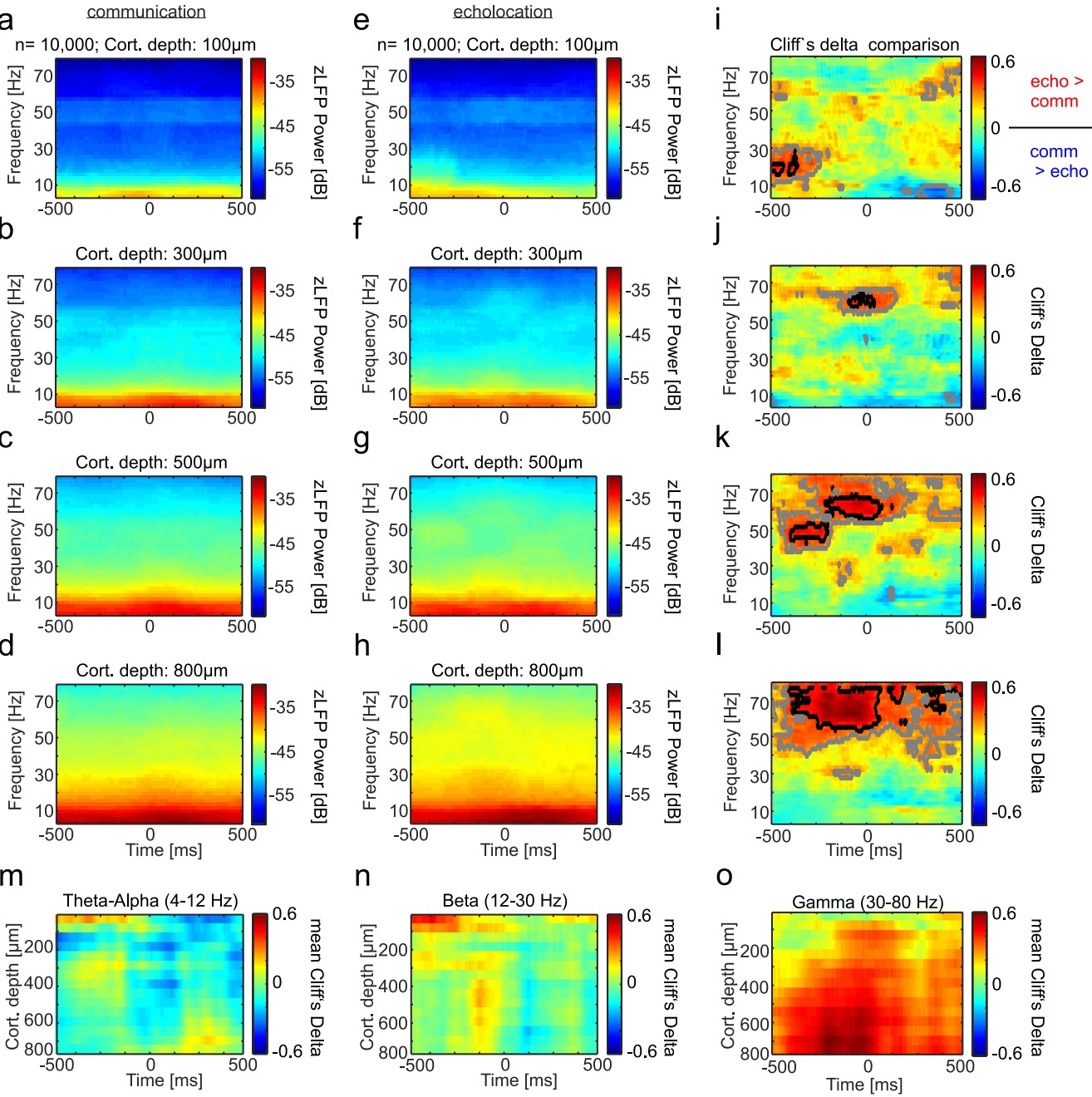

**Fig 4. Time-frequency differences in power distributions across FAF channels, depending on the vocalization type.** **(a)**–**(d)** LFP spectrograms of four illustrative channels of the FAF for the communication condition ($n = 10,000$ randomization trials, see Methods). **(e)**–**(h)** Spectrograms obtained in the same four example channels during echolocation. **(i)**–**(l)** Colormaps of Cliff's Delta values obtained when comparing the time-frequency dynamics in the echolocation and communication conditions in the four example channels. Black highlighted regions indicate large effect size ($d > 0.47$). Gray indicates medium effect size ($d > 0.33$) [60]. **(m)**–**(o)** Mean Cliff's Delta values across FAF depths. Mean values were obtained for all the frequencies that composed the theta (4–8 Hz), beta (12–30 Hz), and gamma (30–80 Hz) bands, represented in panels m, n, and o, respectively. This figure was created based on data that can be found at https://doi.org/10.12751/g-node.6a0d94. FAF, frontal auditory field; LFP, local field potential.

depths studied. Other large spectral differences were found in the theta-alpha range both before and after vocalization with a time- and depth-dependent pattern (see red and blue regions in example channels in Fig 4I–4L and across-depths data in Fig 4M). Differences in

the beta band were pronounced mostly before sound production and occurred at different time points before call onset across cortical depths ([Fig 4N]). Overall, these results suggest that different neural frequency channels in the FAF and CN correlate differently with the bats' vocal output.

We considered the possibility that differences in neural oscillations observed may be directly related to specializations for producing high- versus low-frequency sounds, and only secondarily to the fact that one vocalization set is used for navigation, while the other is used for social communication. To test this possibility, spectral analyses were performed comparing LFPs recorded during the emission of communication calls with low- and high-frequency components (LHF calls) versus echolocation calls (see [Fig 5]). As mentioned in the preceding text, besides high power in low frequencies, LHF calls exhibited pronounced power at frequencies above 50 kHz (see [S2 Fig] and [Fig 5A]). The results of comparing neural activity related to the production of LHF calls versus echolocation pulses are shown in [Fig 5B, 5C–5F] for the CN and FAF, respectively. Overall, the results obtained when considering only LHF calls did not

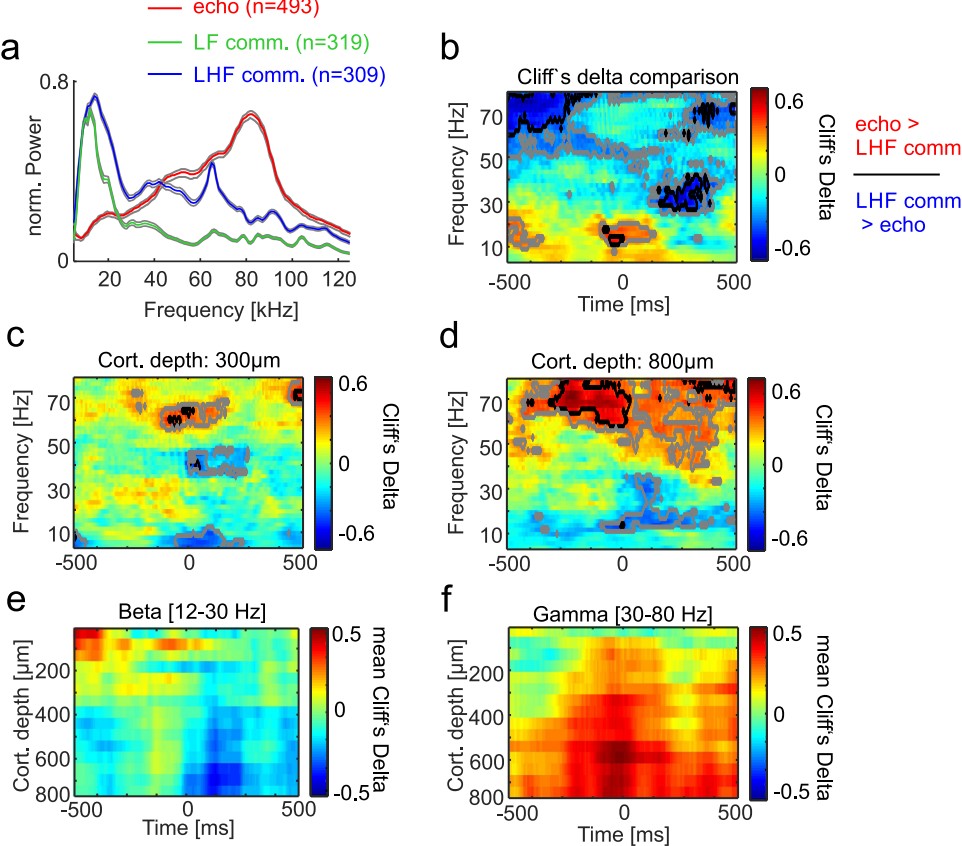

**Fig 5. LFP power differences during the production of LHF communication calls and echolocation pulses.** LHF communication calls carry pronounced energy at both low (<50 kHz) and high frequency (see median vocalization spectra in (**a**)). LF calls carry power only at low frequencies, while echolocation pulses carry power at high frequencies. (**b**) Cliff's Delta effect size measures obtained in the CN when comparing LHF versus echolocation sounds. (**c**)–(**d**) Similar to panel (b), but for the FAF at 300-μm and 800-μm depths, respectively. (**e**)–(**f**) Average Cliff's Delta across FAF depths in the beta and gamma ranges, respectively. Overall, the results obtained when comparing neural activity related to LHF and echolocation call production resembled those obtained when pooling data from all communication calls (see [Fig 4]; for results of comparing LHF and LF calls, see [S6 Fig]). Data underlying this figure can be found at https://doi.org/10.12751/g-node.6a0d94. CN, caudate nucleus; FAF, frontal auditory field; LF, low-frequency; LFP, local field potential; LHF, low- and high-frequency.

differ from those obtained after pooling all communication calls together (compare results presented in Fig 4 and Fig 5). In both cases, differences between echolocation pulse and communication call production appeared before call production in the gamma and beta bands. Comparing LHF and LF communication calls with each other rendered only post-vocalization differences in the gamma range and differences in the theta-alpha range in the FAF localized around the time of call production (see S6 Fig). Taken together, our results indicate that differences in LFP spectral power are not related solely to the presence/absence of high-frequency components in the calls emitted.

## The spectral structure of LFPs predicts vocal output

We used binary support vector machine (SVM) classifiers to assess whether models could be constructed to "predict" the bats' vocal output based solely on the power distribution of LFPs before (or after) call production. SVM classifiers were trained (only once) with 10,000 randomly chosen power distributions across time and frequency bands (for each frequency band, the average power at each time point was calculated; 5,000 randomization trials per call type). In a first analysis step, only spectral power occurring before call onset was considered for training and predicting vocal output. The remaining 10,000 power distributions (5,000 per call type) were used to compute the percentage of correct hits by the models (Fig 6A).

In the CN, when using only pre-vocalization information, low-beta and high-gamma band LFPs provided the best predictions about the type of upcoming vocal outputs (approximately 65% correct hits in both cases; Fig 6A top panel). Note that these frequency bands showed the highest differences in power when comparing both vocalization conditions (cf. Fig 3C). Overall, the FAF provided higher prediction accuracy than the CN (Fig 6A bottom panel). Here, the gamma band (in particular the high gamma band [50–80 Hz]) displayed high accuracy in predicting the type of vocal output, reaching values of approximately 80% accuracy at depths >500 μm. Gamma signals in the FAF also produced the lowest model cross-validation errors (see S7A and S7B Fig). In both brain structures, training the SVM classifiers with false information created by randomization of the labels in training signals led to a drop in prediction capability, with true detection rates around chance level (i.e., 50%, see S7C and S7D Fig).

Next, the same SVM classifier analysis was performed based on the power of LFPs recorded after vocal production (Fig 6B). For the post-vocalization LFP power, the best prediction occurred again in deep layers of the FAF in gamma frequencies (maximum of approximately 78% correct hits). Interestingly, prediction power in the CN was lower in post-vocalization signals when compared to the pre-vocalization time (compare colors in top panels of Fig 6A and 6B). The latter suggests that evoked responses following vocal production in the striatum are poorly correlated with the type of vocalization perceived by the bats.

In a last step, we ran the classifier analysis using a third dataset composed of pre-vocalization LFPs in trials in which the post-vocalization time was contaminated with other sounds produced by the animal. This includes cases in which trains of vocalizations (sometimes mixtures of echolocation and communication) were produced. Despite this possible confound, the classifier was still able to reach accuracy levels of approximately 68% correct hits when considering gamma activity in deep FAF channels (Fig 6C).

## Fronto-striatal coupling occurs in low-frequency bands of the LFP

To investigate the functional coupling between the FAF and CN during vocalization, the neural coherency was calculated. Coherency refers to the trial-averaged cross-spectral density of two signals measured simultaneously, taking into account the phase synchrony of the signals. Here, the magnitude of coherency (defined as "coherence") was calculated between neural

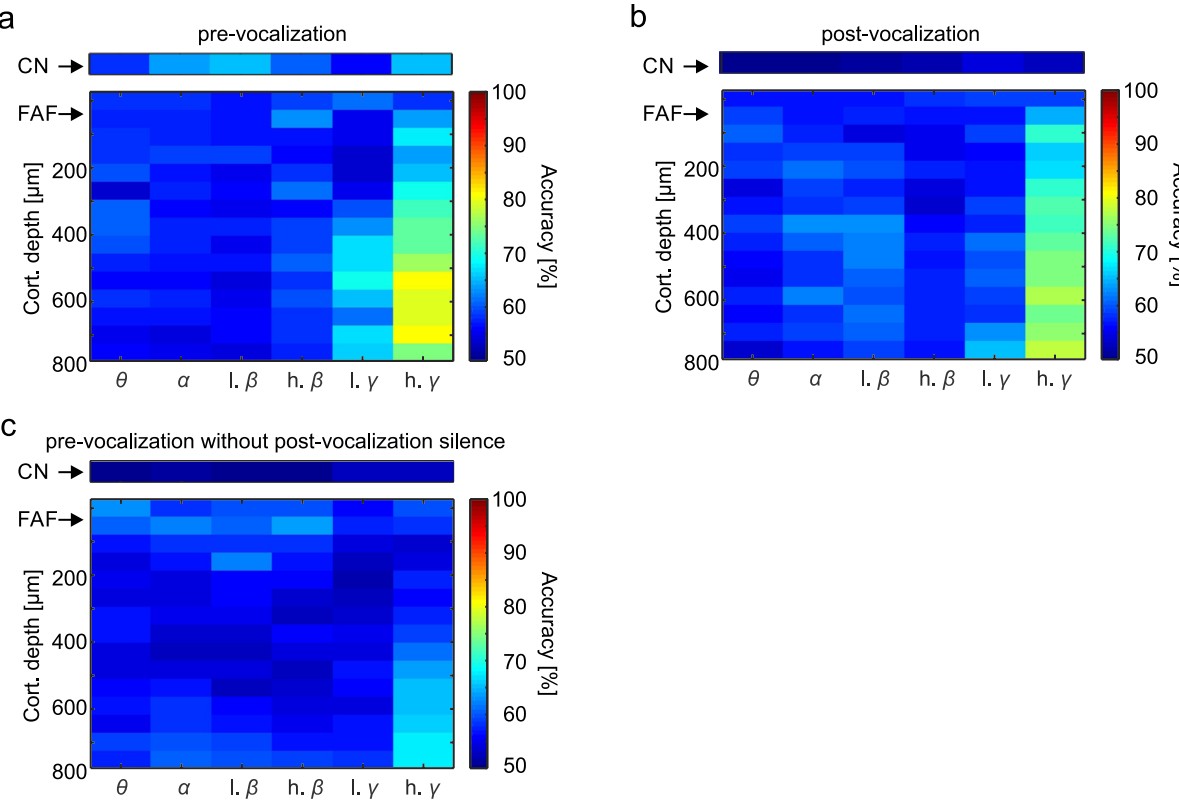

**Fig 6. LFP signals leading to vocalization can be used to predict vocal output. (a)** Prediction accuracy calculated using a binary SVM classifier (see Methods) trained with LFP information (filtered by frequency band) occurring before vocalization in the echolocation and communication conditions (all communication calls were pooled together). Models were trained with half of the data (*n* = 5,000 randomization trials in each vocalization condition). The other data half was used for calculating the models' prediction accuracy. **(b)** Same as panel (a), but in this case the models had to classify post-vocalization activity. Note that in the post-vocalization condition, prediction accuracy dropped in the striatum. In the FAF, accuracy was still highest in deep layers in the gamma range. **(c)** Same as (a) and (b), but here the model had to predict a third dataset corresponding to pre-vocalization activity in trials with contaminated post-vocalization time (training set was the same as in (a)). Even in this case, FAF signals rendered good predictions about ensuing vocal output. The SVM was computed based on data that can be found at https://doi.org/10.12751/g-node.6a0d94. CN, caudate nucleus; FAF, frontal auditory field; LFP, local field potential; SVM, support vector machine.

signals recorded at different depths of the FAF and the CN (Fig 7). The preferred frequencies for coherence between both structures were located in the low spectral range (under 12 Hz, mostly in theta, see below) for both types of vocalizations. There was a striking difference in the temporal pattern of coherence observed in both vocalization conditions. For communication calls, the highest fronto-striatal coherence was found before and during call production (Fig 7A–7D). This temporal pattern could be further divided into LHF communication calls showing highest coherence before call onset and LF communication calls exhibiting coherence maxima during/slightly after call emission (see S8 Fig for coherence patterns for LF and LHF calls). However, when echolocation pulses were produced, coherence shifted to even later time points after call emission (Fig 7G–7J).

The different temporal coherence patterns in the two vocalization conditions were also clear in average coherence plots that display the mean theta and alpha coherence across all FAF depths studied (Fig 7E, 7F, 7L and 7L). Note that regardless of the vocalization type produced, FAF depths below 600 μm rendered the lowest coherence values, even though they displayed the strongest LFP deflections during call production (compare results in Fig 7E and 7K with Fig 2C and 2D). Also note that the gamma band of the LFP was not involved in inter-

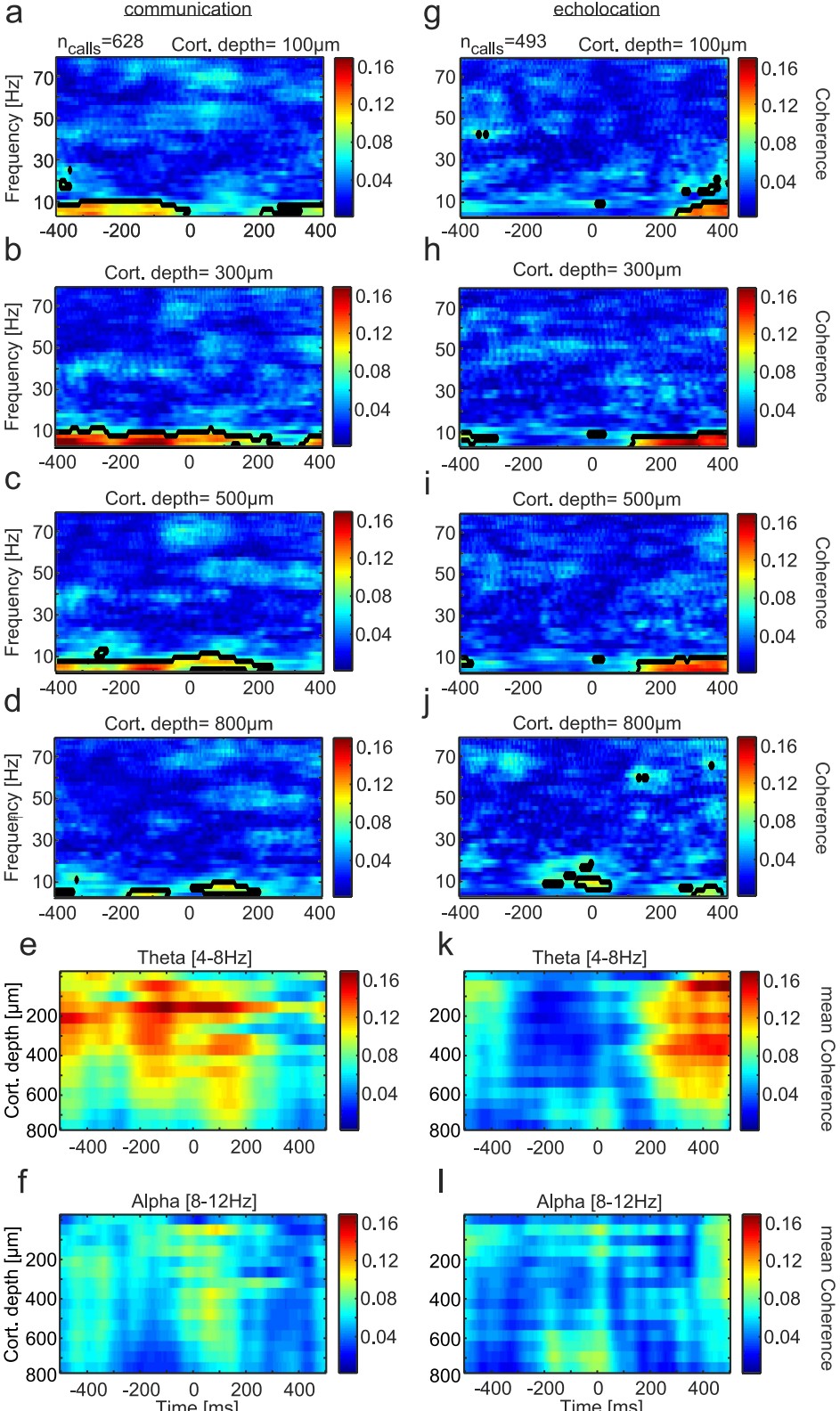

**Fig 7. Functional coupling between the FAF and the CN during vocalization.** Time-frequency resolved coherence between the striatum and four exemplary channels of the FAF at **(a)** 100-μm; **(b)** 300-μm; **(c)** 500-μm; and **(d)** 800-μm depths during communication (*n* = 628 trials). Black lined regions refer to the 95th percentile of all computed

coherence values during vocalization. (**e**) Time resolved coherence strength between both structures across cortical depths in theta and alpha (panel (**f**)) during the production of communication calls. Mean coherence values across frequencies in each range were calculated. (**g**)–(**j**) Coherograms in four example channels during echolocation pulse production (*n* = 493 trials). (**k**)–(**l**) Same as (e) and (f) but for the echolocation case. During echolocation production, pronounced coherence in theta in the top-to-middle layers was found 200 ms after call onset. This temporal pattern differs from that observed during the production of communication calls. The coherence was computed based on data that can be found at https://doi.org/10.12751/g-node.6a0d94. CN, caudate nucleus; FAF, frontal auditory field.

areal coherence, even though this band did show differences in within-structure analysis of LFP signals during echolocation pulse and communication call production (see Results presented in Figs 3 and 4). Taken together, our results indicate temporally defined functional coupling of fronto-striatal circuits depending on the type of vocal output produced by bats.

## Frequency-dependent spike-LFP locking prior to vocalization

We also studied the spiking pattern of striatal and FAF neurons and the relation between spiking and LFP phase. Spiking activity was gathered from spike-sorted single units (see Methods). Peri-stimulus time histograms (PSTHs) computed for the CN did not show clear evidence for evoked responses following vocalization in either vocalization condition (Fig 8A and 8B). In the FAF, spiking was strongest in superficial and deep layers and vocalization-triggered spiking was apparent at depths below 600 μm in both vocalization conditions (Fig 8C and 8D).

The locking between spikes and the phase of LFPs occurring before vocalization was studied. Phase-locking values were calculated by linking spike times to the instantaneous phase of each LFP frequency band (see example phase-locking calculations in S9 Fig). The circular distributions of LFP phases at which spiking occurred for each frequency band were compared with random-phase distributions obtained by extracting LFP phases at time points not related to spiking. To get robust circular spike-phase and random-phase distributions, circular distributions were calculated via bootstrapping (see Methods). Differences in vector strength (dVS) between spike-phase and random-phase distributions were calculated to estimate the strength of spike-phase locking (see circular distributions and vector strengths [VS] in S10 Fig). Significance was assessed by comparing VS values obtained across randomization trials for the spike-phase and random-phase conditions (Bonferroni-corrected Wilcoxon rank-sum test $p < 0.001$, see Methods).

In the CN, significant differences between spike-phase and random-phase distributions were found in the theta band during communication and in the alpha and high-gamma bands during echolocation (Fig 9A and 9B). When comparing VS distributions from both vocalization conditions (not with the surrogate data) in the striatum, significant differences were only found in the high beta range (Fig 9C). The FAF showed statistically significant spike-phase locking in several LFP frequency bands and cortical depths (Fig 9D and 9E). In particular, spike-phase locking in the low- and high-gamma LFP bands was pronounced across layers, and consistent differences appeared when comparing between vocalization types in the low-gamma range at FAF depths >600 μm (Fig 9F). Besides the gamma spike-phase locking observed, in the communication condition, there was consistent spike-phase locking in the theta band at depths spanning from 250 to 400 μm (Fig 9D), although this effect was not significant when comparing between vocalization types (Fig 9F). Note that we refer to "consistent" spike-phase locking differences whenever statistical significance occurred in more than two contiguous FAF channels. The effect size calculations (e.g., Cliff's Delta) complementing rank-sum testing rendered in all cases values below 0.2, thus indicating high data variability (see effect size plots in S11 Fig). Overall, our results indicate that coupling between LFPs and spiking occurs in the striatum and deep layers of the FAF before vocal production.

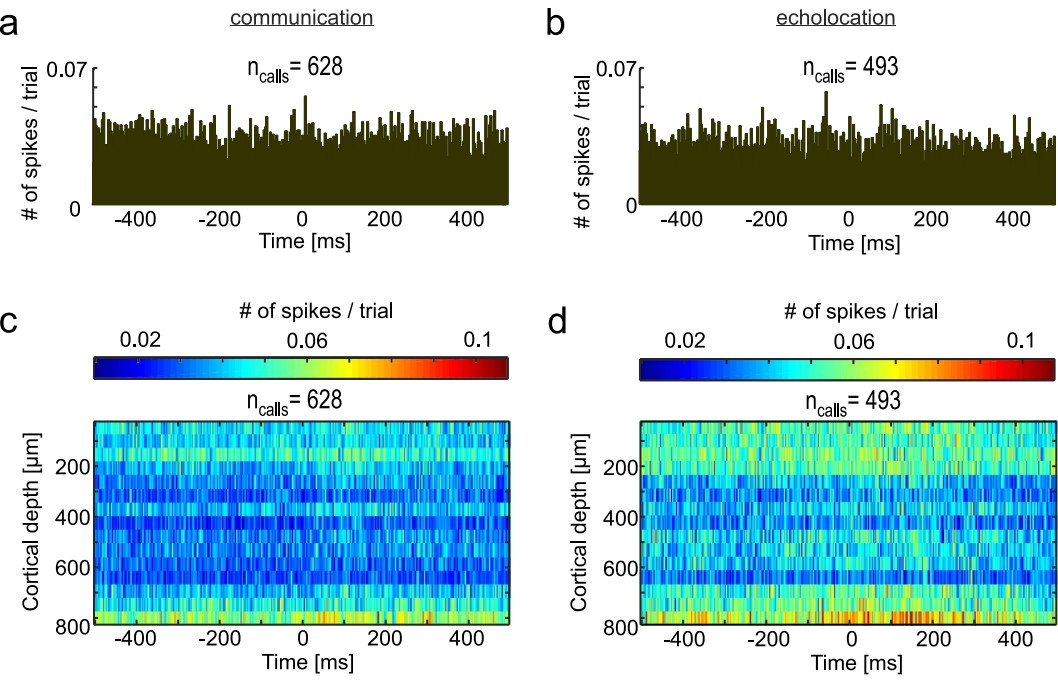

**Fig 8. Spiking activity in the CN and FAF during vocalization.** Spiking probability (computed as numbers of spikes per trial per bin, binsize = 3 ms) in the CN 500 ms before and after communication ((**a**), $n$ = 628 trials) and echolocation ((**b**), $n$ = 493). Spiking across all channels recorded in the FAF during communication (**c**) and echolocation (**d**). In the FAF, during both types of vocalizations, distinct spiking activity could be identified in deep layers. One reason for the small increase in spiking activity in response to the vocalization could be due to the sparse distribution of vocalization relevant neurons in the FAF. Data underlying this figure can be found at https://doi.org/10.12751/g-node.6a0d94. CN, caudate nucleus; FAF, frontal auditory field.

## Discussion

Previous work has shown alterations in the fronto-striatal network in disorders with impaired speech production in humans [8,61]. However, electrophysiological mechanisms by which fronto-striatal activity could participate in vocal production in humans and other vertebrate species remain elusive. In this article, we show that neural oscillations in fronto-striatal circuits are distinctly linked to the type of vocalizations produced by bats. The main findings reported in this paper include the following:

1. A unique intra-areal pattern of LFP frequency representation during vocalization (most prominent in beta and gamma LFP ranges (12–30 and 30–80 Hz, respectively), which can be used to predict ensuing vocal actions.

2. Functional coupling between the CN and FAF in low frequencies (theta, 4–8 Hz) with temporally distinct characteristics depending on the vocal output.

3. The occurrence of spike-LFP phase locking, especially in frontal areas in the gamma LFP band prior to vocalization.

Taken together, these results suggest a functional involvement of the fronto-striatal network in neural processing for selecting and producing different types of vocalizations, with the capacity to discriminate between, and predict, different vocal motor outputs. Moreover, neural activity in the FAF and CN appears to correlate on a LFP and single-unit basis to vocalization, but appears to be coupled in distinct frequencies and time points in relation to the vocal motor

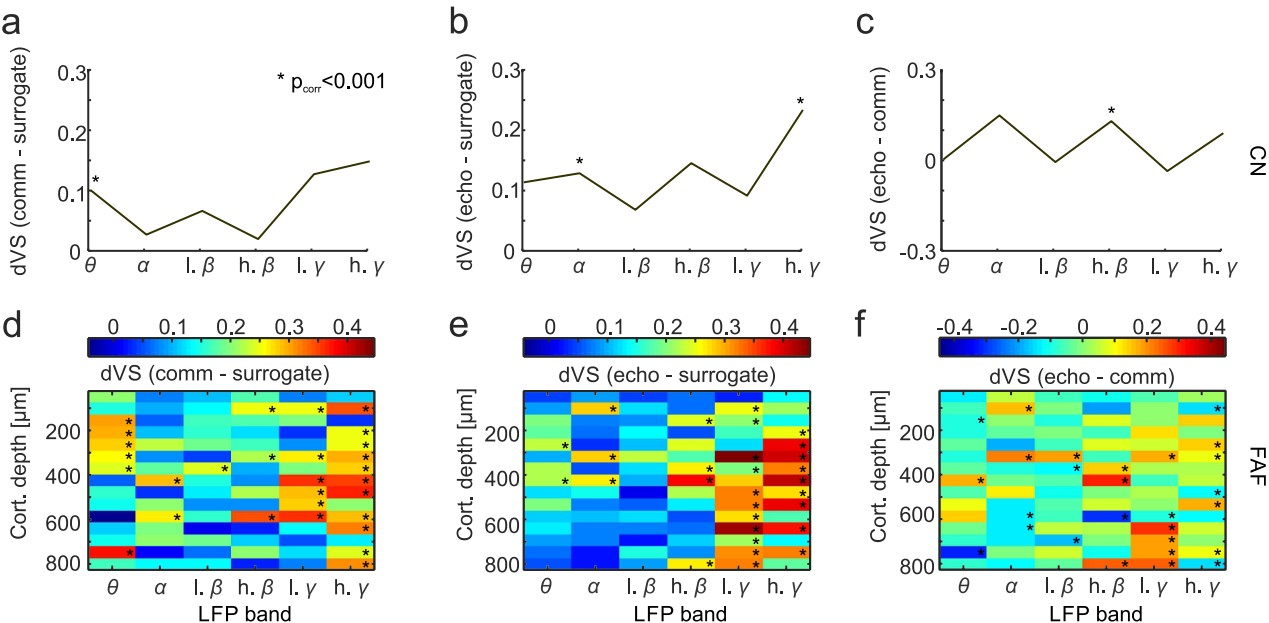

**Fig 9. Spike-phase locking across vocalization conditions. (a)** dVS obtained before vocalization onset in the communication-surrogate condition in the CN; **(b)** echolocation-surrogate condition; and **(c)** echolocation-communication condition. **(d)**–**(f)** dVS values computed for all recorded channels in the FAF in the three conditions mentioned above. Statistical differences were tested by comparing VS distributions (Wilcoxon rank-sum tests with Bonferroni correction, $^*p < 0.001$, see Methods). Data underlying this figure can be found at https://doi.org/10.12751/g-node.6a0d94. CN, caudate nucleus; dVS, difference in vector strength; FAF, frontal auditory field; VS, vector strength.

action. A graphical abstract summarizing the results presented in this manuscript can be found in Fig 10 (see also the summary presented in S1 Table).

## Linking fronto-striatal oscillations to vocal output: General considerations

Our hypothesis that echolocation pulses and communication calls involve different fronto-striatal network dynamics could be corroborated. In bats, differences related to the production

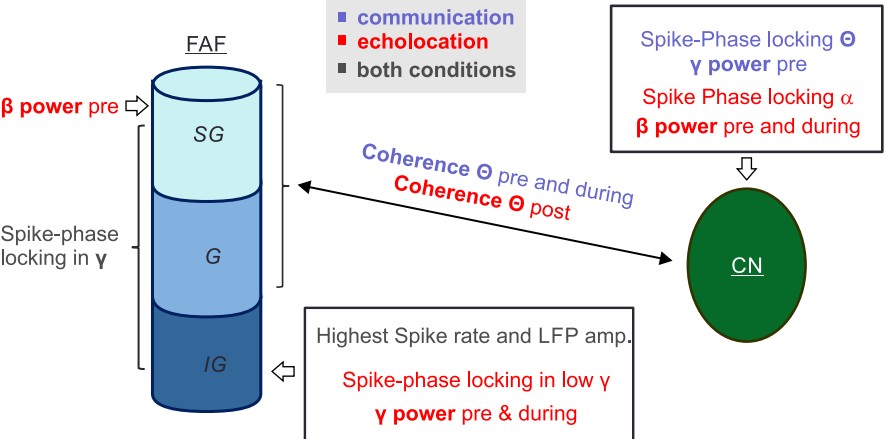

**Fig 10. Visual abstract depicting the main results presented in this study.** "SG," "G," and "IG" indicate a putative subdivision of the FAF into supragranular, granular, and infragranular layers, respectively. Note that we do not report data on the directionality of the connection between both regions, and thus functional coupling is displayed with a double arrow. Different electrophysiological parameters such as LFP power measurements, spike-phase locking and LFP–LFP coherence demonstrate that fronto-striatal circuits can predict ensuing vocal output in bats. CN, caudate nucleus; FAF, frontal auditory field; LFP, local field potential.

and processing of different types of vocalizations encompass neural oscillations in the theta, beta, and gamma bands. This study presents only correlative evidence linking vocal production to neural oscillations in the fronto-striatal network. The latter does not imply a causal role of neural oscillations in vocal control.

Neural oscillations are a generalized phenomenon in the nervous system, and they have been studied extensively in past years (for reviews see [29,35,62–64]). The current consensus is that oscillations represent different excitability states in neural populations. Oscillatory activity differs across brain structures and it participates in processes such as inter-areal synchronization, local information binding, selective attention, and memory formation, among others [27,65,66]. The neural mechanisms by which oscillations are generated in the brain are still under debate. At least gamma oscillations observed in the neocortex and hippocampus seem to originate from an interplay between excitatory and inhibitory activity in pyramidal cells and interneurons ("pyramidal-interneuron gamma" [PING] networks, for review see [62,67]). It has been postulated that oscillations in other frequency bands could also be linked to pyramidal-interneuron networks (PIN) that oscillate with different time constants (see for example the PIN-theta networks proposed for the auditory cortex [64]). Low-frequency oscillations (i.e., alpha and theta) also have been linked to pace-making pyramidal neurons, although it is not clear if the underlying mechanism for pace-making relates to pyramidal-interneuron networks as well [68,69]. In humans, beta oscillations found in the striatum also seem to involve inhibitory interactions between neurons [70]. With our current data, we cannot assess the cellular mechanisms responsible for the oscillations observed in the bat striatum and frontal cortex during vocalization. Regardless of the cellular origin of fronto-striatal oscillations, our data show that, at least in bats, neural rhythms in these two structures correlate well with vocal output.

In the present study, we focused on analyzing oscillatory activity related to vocalizations that were surrounded by silent periods to avoid possible confounds related to the production of call trains (except in Fig 6C, where calls with contaminated post-vocalization times were used as control for the prediction analysis). We reasoned that situations in which bats produced vocalization trains with mixtures of echolocation and communication calls could render misleading results. Future studies could assess whether pre-vocalization activity carries information about the physical parameters of vocalization trains.

Overall, the observed electrophysiological effects during communication call production need to be considered cautiously. Bats produce communication calls in numerous situations such as, e.g., distress calling, courtship, and territorial disputes, among others [44,45]. The most common way to parse communication calls into different subcategories is to score the behavior/context during which the calls are produced [59,71,72]. As we recorded in head-fixed animals, it is difficult to assess what type of communication calls were broadcasted. Pooling vocalization trials from many types of communication calls together could potentially hide call-specific effects.

Based on spectro-temporal features, we identified two types of communication vocalizations (containing either only low-frequency or high- and low-frequency components, see Fig 5 and S2 Fig). Even communication calls containing pronounced power at high frequency differed markedly from echolocation vocalizations. The latter indicates that differences observed in terms of LFP spectral structure and inter-areal coupling are not related solely to the absence/presence of high frequencies in the sounds uttered. Note that we cannot discard that communication calls containing only high frequencies involve LFP patterns similar to those observed during echolocation. Communication calls carrying only high frequencies were not observed in our dataset, but *C. perspicillata* produces such calls in contexts such as mating [59].

## Theta oscillations for inter-areal coupling in the fronto-striatal circuit

According to our data, in bats, vocal production is linked to activity in the theta, beta, and gamma bands of LFPs. The production of communication calls and echolocation pulses rendered power in the theta range after vocalization (cf. Fig 4A with Fig 4B, across-depths data in Fig 4M) as well as vocalization dependent inter-areal coherence patterns (Fig 7). Power differences after call onset also occurred in the beta and gamma range (Fig 4N and 4O). The time period after vocal production must be examined carefully, as the calls produced could differ in their acoustic attributes (i.e., frequency composition and duration, among others), which could lead to differences in call-evoked neural responses. In different sensory cortices, low frequencies such as theta and alpha are known to modulate sensory processing and enable sensory selection [29,73,74]. Unlike sensory cortices, low-frequency oscillations in frontal areas are less understood in terms of sensory processing.

Our data suggest that low-frequency rhythms in frontal areas (especially theta, see Fig 7) relate to inter-areal communication between FAF superficial layers and the dorsal striatum during vocalization, as quantified by computing inter-areal LFP coherence (Fig 7). This result falls in line with a putative involvement of low-frequency oscillations in long-range synchrony [27,28]. The FAF constitutes a nonclassical sensory area, and its laminar structure (i.e., location of inputs and outputs, such as layer 4 and 5 in sensory cortices [75]) needs further anatomical exploration.

According to our data, FAF layers could hold a crucial role in oscillatory communication between frontal and striatal regions during vocalization initiation. When assessing the coupling between fronto-striatal regions, the timing of inter-areal coherence seems to play an important role when planning and producing different types of sounds (see Fig 7 and S8 Fig). While the highest level of coherence was found before, during, or shortly after (<250 ms) communication call production, echolocation-related coherence occurred at least 250 ms after pulse onset. One possible explanation for the strong coherence following echolocation pulses could be that the latter require a more thorough sensory processing and auditory feedback after vocal production (i.e., for echo evaluation) than communication calls. Such post-processing of echolocation pulses could be relevant for planning ensuing vocal actions and for a coherent representation of the environment in bats. Note that the inter-areal coherence results presented in this paper have implications beyond bat echolocation, as they suggest that temporally precise oscillatory coupling in the fronto-striatal circuit correlates with the production of different vocal outputs. Such fine communication synchrony between brain structures could be affected in conditions such as Parkinson, Huntington disease, and Asperger syndrome, in which fronto-striatal impairments have been described [8,61].

## Intra-areal beta and gamma oscillations provide neural correlates of vocal output

Differences in LFP activity preceding vocal production also occurred in the beta band. According to our data in bats, the beta band of LFPs is differentially involved in echolocation pulse and communication call emission. Beta power is highest during echolocation production in the CN and in superficial layers of the FAF (see Fig 3 and Fig 4). As especially the beta band activity is correlated with motor action planning and performance [36,37], one could hypothesize that the strength of beta oscillations in the CN and in superficial FAF layers is linked to different sensorimotor programs required for the production and/or post-vocalization evaluation of acoustic signals (i.e., echoes during echolocation). Overall, beta is typically dominant in the motor system, correlating with the maintenance of ongoing sensorimotor cognitive states and endogenous timing processes [35,76]. Aberrant beta oscillations (especially in the striatum)

are also a key feature of the parkinsonian brain [4,8,77]. Our results together with those from previous studies suggest that beta oscillations within the fronto-striatal path are important for vocal motor output production.

Another large vocalization type–dependent effect was detected in the gamma band. Before echolocation, high power in this frequency band was observed in deep layers of the FAF, whereas before communication, high gamma power was found in the CN. As the power maxima in gamma was reversed across vocalization conditions in both brain structures, it could be suggested that each component of the fronto-striatal path relies on a differential power distribution of high frequencies in order to produce the same vocal output. This could be supported by the fact that in both brain structures, power in the gamma band was the best predictor of vocal output (Fig 6).

Gamma LFPs also appear to be related to spiking activity. The time periods before both echolocation pulses and communication calls displayed significant phase-locking values in the gamma range across FAF layers (Fig 9D and 9F). The latter suggests a generalized role of gamma-phase coupling preceding vocalization. Spike-phase locking in the gamma range has been demonstrated previously, correlated to vocalization in the sensorimotor nucleus of zebra finches [24].

Classical functions of gamma rhythms across species are linked to selective attention, cortical computation, and working memory [25]. In bats, changes in gamma power have been associated with the processing of auditory stimulation in the bat auditory cortex and with social interaction in frontal areas [78,79]. Moreover, an increase in gamma power was found in the superior colliculus after the production of clusters of echolocation pulses in freely flying bats [48]. The latter could relate to the detected rise of gamma power before echolocation in comparison to communication in the FAF (this study), and could indicate the putative importance of gamma rhythms during navigation, whether the animals are freely flying (as in previous studies) or exploring their environment using their biosonar from a fixed location (this study). Note that gamma oscillations were not involved in long-range fronto-striatal communication (see Fig 7). This finding supports the current view of gamma rhythms being important for local neural computations [25,27].

To our knowledge, changes in gamma power linked to a specific motor action have not been described before for the CN. The ventral striatum is known to display a prominent pattern of gamma power during reward ingestion or decision-making [80], but the oscillatory properties of the nuclei that form the dorsal striatum (such as the CN) are less studied. Our results show that not only the FAF but also the gamma power in the CN are correlated with the type of vocal output.

Taken together, the findings presented in this manuscript indicate that neural oscillations in the gamma and beta bands in fronto-striatal brain regions represent ensuing vocal actions in bats, while oscillations in the theta-alpha range represent the differential sensory processing of the type of call uttered and play a role in long-range inter-areal coupling. Our data suggest that fronto-striatal circuits are an important component of canonical networks underlying vocalization in mammals, and that these circuits could bear key specializations supporting bat echolocation.

## Methods

### Ethics statement

All experiments described in this article comply with current guidelines and regulations for animal experimentation and the Declaration of Helsinki. Experiments were approved by the Regierungspräsidium Darmstadt, Germany (permit number: FU1126).

## Surgical procedure

For neurophysiological recordings, 4 adult Seba's short-tailed bats (*C. perspicillata*) were used. The animals originated from a breeding colony at the Institute for Cell Biology and Neuroscience, Goethe University, Frankfurt am Main (Germany). Bats underwent a surgical procedure for gaining access to frontal and striatal brain regions. The surgery encompassed the implantation of a chronic tetrode mounted on a microdrive in the striatum, a craniotomy above the FAF for the insertion of a linear silicon probe, and the attachment of a custom-made metal rod. The latter facilitated stable recording conditions by preventing head movements. The implantation protocol was modified from the procedure used in previous studies [73,74,81–84].

First, after monitoring the health status, bats were anesthetized subcutaneously with a mixture of ketamine (10 mg/kg Ketavet, Pfizer, Berlin, Germany) and xylazine (38 mg/kg Rompun, Bayer, Leverkusen, Germany) and topically with local anaesthesia (Ropivacaine 1%, AstraZeneca GmbH, Wedel, Germany). After achieving stable anaesthesia conditions, animals were placed on a heating blanket (Harvard Apparatus, Homoeothermic blanket control unit, Holliston, MA) at 28°C. Afterwards, the fur on top of the head was excised, the skull was exposed via a longitudinal midline incision, and the skin, connective tissue, muscle, and debris were removed. Using macroscopically visible landmarks (e.g., the pseudocentral sulcus and blood vessels), the skull was evenly aligned. With a scalpel blade, a first craniotomy (approximate 2-mm diameter) was made between the sulcus anterior and pseudocentral sulcus for the chronic implantation of a tetrode (Q1-4-5mm-200-177-HQ4_21mm, NeuroNexus, Ann Arbor, MI, see S1 Fig) mounted on a moveable microdrive (dDrive-m, NeuroNexus, Ann Arbor, MI) to ensure mobility of the electrodes. To prevent the electrodes from bending, the tetrode was introduced into the tissue (partial implant with 2.0-mm depth) with an angle of 17° perpendicular to the brain surface under the microscope. Subsequently, the microdrive was fixed to the scalp with a two-component UV-acrylic glue (Kulzer GmbH, Hanau, Germany) and dental cement (Paladur, Kulzer GmbH, Hanau, Germany) and was placed via a screw (1 full counterclockwise turn = 150 μm) at the target position (in total: >2.1-mm depth). For protection and shielding, a plastic cap covering the implant was glued using UV-acrylic. The connector was permanently attached to the cap. For stability purposes, a custom-made metal rod (2-cm length, 0.1-cm diameter) was fixed to the surface of the bat's skull. The metal post was glued using UV-acrylic and dental cement to the bone and the plastic cap posterior to the tetrode (see sketch in S1A Fig). All efforts were made to reduce the weight of the implant and the bat health status was carefully monitored throughout the experiments.

Before starting the recordings, a second craniotomy (2–3-mm diameter) rostral to the tetrode between the sulcus anterior and longitudinal fissure above the FAF was implemented using a scalpel blade [52]. To record extracellular action potentials and LFPs in the FAF, an acute A16 laminar probe (NeuroNexus, Ann Arbor, MI, S1B Fig) was introduced into the brain on each recording day. After surgery, the animals had at least 48 hours of recovery before starting electrophysiological recordings.

## Neurophysiological recordings in vocalizing animals

All experiments were performed chronically for a maximum of 2 weeks after surgery. Whenever the wounds were handled, local anaesthesia (Ropivacaine 1%, AstraZeneca GmbH, Wedel, Germany) was administered topically. Before starting the electrophysiological recordings, the bat was placed in a custom-made holder with an attached heating blanket (see previous section) in a Faraday chamber. Subsequently, the tetrode was connected via an adaptor (Adpt. CQ4-Omnetics16, NeuroNexus, Ann Arbor, MI) to a micro amplifier (MPA 16,

Multichannel Systems MCS GmbH, Reutlingen, Germany). For detecting neural activity in the FAF, the laminar probe was lowered through the craniotomy under the cortical surface using a micro manipulator (piezo manipulator PM101, Science Products GmbH, Hofheim, Germany) with a speed of 50 μm/s. The linear probe spanned cortical depths of 50–800 μm below the brain's surface, with channels evenly distributed in 50-μm steps. One silver wire was placed above the dura mater through a third small craniotomy and served as common ground electrode for both the tetrode and the laminar probe. The reference of each electrode array was short-circuited with the respective top recording channel (the electrode closest to the brain surface) to obtain local signals and prevent movement artifacts. Neuronal signals from the striatum and FAF were preamplified and connected via flexible cables to a portable multichannel recording system with integrated AD converter (Multi Channel Systems MCS GmbH, model ME32 System, Reutlingen, Germany). The recording was digitized at a sampling frequency of 20 kHz (16-bit precision). For monitoring, visualizing, and storing the data, MC_Rack_Software Version 4.6.2 (Multi Channel Systems MCS GmbH, Reutlingen, Germany) was used.

For the acquisition of vocal outputs, a microphone (CMPA microphone, Avisoft Bioacoustics, Glienicke, Germany) was placed 10 cm in front of the animal. Acoustic recordings were conducted with a sampling rate of 250 kHz. Vocalizations were amplified (gain = 0.5, Avisoft UltraSoundGate 116Hm mobile recording interface system, Glienicke, Germany) and stored in a PC using the Avisoft Recorder Software (Avisoft Bioacoustics, Glienicke, Germany) with 16-bit precision. Offline analysis was conducted to separate vocalizations into echolocation and communication calls based on their spectro-temporal structure.

In order to synchronize the recording of the vocalization signals and the neurophysiological signals, Matlab-generated triggers (i.e., a sound for acoustic recordings and a TTL pulse for the neural acquisition system) were used to align both recordings. Each recording comprised 3 × 10-minute vocalization experiments, during which bats were let to vocalize at their own volition, with a short break to stimulate vocal production by opening and closing the recording chamber.

## Acoustic stimulation

To estimate the responsiveness of the areas studied to acoustic stimuli, a frequency tuning paradigm was used. Frequency tuning was controlled via a custom-written Matlab software (Math Works, Natick, MA). A stimulation speaker (NeoCD 1.0 Ribbon Tweeter; Fuontek Electronics, Jiaxing, China) was placed 12 cm in front of the animal and pure tones were presented ranging from 10 to 90 kHz in 5-kHz steps (randomized order, repetitions of each pure tone = 30 times) for a duration of 10 ms (0.5-ms rise/fall time) at 60 dB SPL. Following digital-to-analogue conversion using a soundcard (RME Fireface 400, 192 kHz, 24-bit), the generated pure tones were amplified (Rotel power amplifier, model RB-1050, Worthing, United Kingdom) and presented to the bats.

## Analysis of LFP data

The analysis was implemented using custom-written Matlab scripts (MATLAB R2015b, The Math Works, Natick , MA). All vocalizations were assessed offline using the Avisoft SAS Lab Pro software (v.5.2 Avisoft Bioacoustics, Glienicke, Germany). The initial acoustic trigger, communication calls (typical power maximum around 5–50 kHz) and echolocation pulses (peaking above 50 kHz, [57,58]) were manually located, individually labelled, and their timing was exported to Matlab. To evade response contamination by other auditory stimuli, the "clean" communication calls and echolocation pulses were identified, which comprised at least 500 ms without any vocalization prior to and following call production. Spectrograms of the

vocalizations were calculated with a frame width of 0.8 ms, a frame shift of 0.05 ms, and a hamming window of 2,048-points length.

The peak frequency of each call was estimated from the de-noised FFT. De-noising was achieved by subtracting the FFT of the noise floor to the FFT of the call in question (duration of call and noise were matched in each case). Communication calls were split into two groups: one group of communication calls with only high power at low frequencies (<50 Hz, LF calls) and a second group that showed pronounced power in low and high frequencies (>50 kHz, LHF calls). LF and LHF communication calls were classified based on their spectrum; e.g., a call was assigned to the LHF group if the power maximum at frequencies above 50 kHz was at least larger than half the power at frequencies below 50 kHz.

To investigate LFPs during each calling condition, the electrophysiological signal was filtered between 1 and 90 Hz (second-order Butterworth filter), the line noise removed using the *rmlinesmovingwinc* function of the Chronux toolbox [85], and down-sampled from 20 kHz to 1 kHz. Additionally, the signals were normalized by calculating the z-score at each time point by subtracting the mean and dividing by the standard deviation per recording. Z-scoring was conducted across channels for the FAF (to keep amplitude relationships across channels) and for each channel of the CN individually.

To extract LFP fluctuations linked to vocalization, a randomization procedure was used. This randomization procedure rendered 10,000 communication and echolocation signals for the CN and each recording channel of the FAF. Each randomization trial was obtained by averaging 100 randomly chosen LFPs corresponding to either the communication or echolocation condition. Note that because of extensive averaging, this randomization procedure removes signal components that are not locked to the vocalizations.

Time-frequency analysis was conducted for each randomization trial using the Chronux function *mtspecgramc* with a 250-ms window size, 0.5-ms time step, and a time-bandwidth product of 2 with 3 tapers. To compute the difference in power during the production of different call types, the logarithmic power spectrogram of the communication condition was subtracted from the logarithm of the power spectrogram obtained during echolocation. Statistical power was evaluated using Cliff's Delta (*d*). This measure ranges between −1 and 1, with almost identical observations rendering *d*-values around zero. The *d*-value borders for defining large, medium, and small effect sizes were set to 0.474, 0.333, and 0.147, respectively [60].

A binary SVM classifier was used for predicting vocal output using the average spectral signal in each LFP band either before or after vocalization. The SVM classifier was trained (*fitcsvm* function, rbf kernel, Matlab 2015, single training, no standardization, fitting posterior probabilities after model creation) using signals obtained in 10,000 randomization trials (5,000 per vocalization type, see preceding text). SVM models obtained were cross-validated using 10-fold cross-validation. In a second step, labels were swapped in the training set before classification to assess the performance of the models in the absence of reliable training information.

To evaluate the oscillatory coherence and phase consistency between signals in the striatum and the different cortical depths of the FAF, the Chronux function *cohgramc* with the same parameters used for spectral analysis (see neural spectrogram specifications above) was used. This operation performed coherency calculations between all possible pairs of different channels in the FAF and each channel in the striatum (here, no randomization was used; in other words, we used the LFPs linked to the production of each echolocation and communication trial). Then, the average coherogram obtained between FAF channels and each striatal channel was calculated. For displaying and assessing the strength of coherency, the magnitude of the coherency ("coherence") was used. Coherence values exceeding the 95th percentile of all coherence values obtained were labelled as significant.

Using the same pre-processing methods described above (filtering, down-sampling, z-scoring per recording, and demeaning), LFP responses obtained from the frequency tuning paradigm were quantified. The absolute value of the analytical signal (obtained after Hilbert transforming) was used to calculate the instantaneous energy of each recording channel in response to each sound frequency tested. The frequency eliciting the highest amount of energy was labelled as best frequency.

## Analysis of spike data

Spiking activity was acquired by filtering neural signals in the frequency range of 300–3,000 Hz (second-order Butterworth filter). Spike detection was performed using the SpyKING CIRCUS toolbox with automatic clustering and a threshold of 5 median absolute deviations using the best spiking template per channel and recording [86]. With a bin size of 3 ms, PSTHs were calculated for both brain structures.

To investigate the relationship between spikes and LFPs, phase-locking values were calculated using the circular statistics toolbox [87]. For phase-locking calculations, only the time window before vocalization was considered. The procedure used for calculating phase locking values is illustrated in S9 Fig for one example echolocation trial. After extracting spike times and raw LFPs related to the isolated vocalization trial, the LFP signal was filtered in different frequency bands (e.g., theta [4–8 Hz], alpha [8–12 Hz], low beta [12–20 Hz], high beta [20–30 Hz], low gamma [30–50 Hz], and high gamma [50–80 Hz]). Filtered LFPs were Hilbert-transformed, and their instantaneous phase information was extracted. The phase at which spiking occurred for each LFP frequency band was stored and analyzed using circular statistics (see below).

Circular distributions of LFP phases at which spiking occurred for each frequency band were compared with random-phase distributions obtained by extracting LFP phases at random time points not related to spiking. To get robust circular spike-phase and random-phase distributions, circular distributions were calculated 10,000 times, with 100 randomly chosen spike-phase and random-phase values included in each randomization trial. Two parameters were extracted from the circular distributions obtained in each spike- and random-phase trial: the distribution's VS (*circ_r* function in the circular statistics toolbox [87]) and its angular mean (*circ_mean* function in the circular statistics toolbox [87]). VS values obtained from all randomization trials were used for assessing statistical significance when comparing spike-phase and random-phase distributions using Bonferroni-corrected Wilcoxon rank-sum tests ($p < 0.001$). Angular mean values were used for visual display and for calculating population VS differences (dVS). In our calculations, positive dVS values indicate higher VS in the spike-phase distribution when compared to the random-phase control.

## Histological verification of striatal recordings

For visualization of the electrode implantation location, histological analysis was performed following the completion of the experiments. To locate the tracks of the chronically implanted tetrode in the striatum, an electric lesion was performed for 10 seconds with 10 μA DC current using a Stimulus Isolator A365 (World Precision Instruments, Friedberg, Germany) under deep anaesthesia prior to perfusion. Electric lesions were set for each animal on the last experimental day on the most ventral and dorsal striatal electrodes. Subsequently, the animals were euthanized with an intraperitoneal injection of 0.1 mL sodium pentobarbital (160 mg/mL, Narcoren, Boehringer-Ingelheim, Ingelheim am Rhein, Germany) and transcardially perfused using a peristaltic pump (Ismatec, Wertheim, Germany) with a pressure rate of 3–4 mL/minutes. The bats were perfused with 0.1 M phosphate buffer saline for 5 minutes, followed by a

4% paraformaldehyde solution for 30 minutes. After removing the surrounding tissue, muscles, and skull, the brain was carefully eviscerated, fixed in 4% paraformaldehyde at 4˚C for at least one night, and placed in an ascending sucrose sequence solution (1 hour in 10%, 2–3 hours in 20%, 1 night in 30%) at 4˚C to avoid the formation of ice crystals in the tissue. Subsequently, the brain was frozen in an egg yolk embedding encompassing the fixation in glutaraldehyde (25%) with $CO_2$. For sectioning the frozen brain, a cryostat (Leica CM 3050S, Leica Microsystem, Wetzlar, Germany) was utilized and coronal slices (50 μm thick) were prepared, mounted on gelatin-coated slides and Nissl stained. In brief, the brain slices were immersed in 96% ethanol overnight and 70% ethanol (5 minutes), hydrated in distilled water (3 × 3 minutes), stained in 0.5% cresylviolet (10 minutes), rinsed in diluted glacial acetic acid (30 seconds), differentiated in 70% ethanol + glacial acetic acid until neuronal somata were still red-violet stained with only faint coloration of the background, fixed in an ascending alcohol sequence (2 × 5 minutes in 96% ethanol, 2 × 5 minutes in 100% isopropyl alcohol), cleaned by Rotihistol I, II, and III solution (Carl-Roth GmbH, Karlsruhe, Germany) and covered with DPX mounting medium. The inspection of the lesion was facilitated by a bright-field, fluorescence microscope (Keyence BZ-9000, Neu-Isenburg, Germany). A Nissl staining of a bat brain with the associated track of a chronically implanted HQ4 tetrode in the dorsal part of the CN can be found in S1D Fig.

## Supporting information

**S1 Fig. Electrode implantation procedure. (a)** Schematic outline of the implantation sites. Olfactory bulb denotes the anterior part of the brain, while the cerebellum is found in the posterior part. **(b)** Mapping of the A16 laminar silicon probe with 50-μm spacing between electrodes (which was implanted in the FAF), and **(c)** the HQ4 laminar tetrode with 200 μm between recording sites chronically implanted in the CN. **(d)** The Nissl -stained section, including the track of an HQ4 laminar tetrode implanted in the CN (4× magnification). CN, caudate nucleus; FAF, frontal auditory field.
(EPS)

**S2 Fig. Parsing communication vocalizations into LHF and LF calls. (a)** Example communication call containing pronounced power at low (i.e., <50 kHz) and high frequencies. **(b)** Example LF call. **(c)** Average spectrum of LHF and LF vocalizations. Data underlying this figure can be found at https://doi.org/10.12751/g-node.6a0d94. LF, low-frequency; LHF, low- and high-frequency.
(EPS)

**S3 Fig. CN and FAF display auditory responsiveness to stimulation with pure tones. (a)** Mean population striatal activity ± SEM in response to the best frequency (bf) across recording sites. The arrow indicates stimulus onset. **(b)** Bitmap of the amplitude of z-scored LFPs from 100 ms before up to 350 ms after the stimulus onset across cortical depths in the FAF. **(c)** Histogram of bfs in the striatum. Mean (M) and SEM are indicated. **(d)** Histogram of bfs in the FAF. Both brain structures exhibited pronounced auditory responsiveness to low frequencies. **(e)** Distribution of the mean correlation coefficient obtained by correlating tuning curves of all simultaneous recordings in different FAF depths. The mean value across recordings ($n = 47$) was calculated. The high mean correlation (0.68) indicates similar tuning in neighboring channels. **(f)** Exemplary area under the curve (AUC) of the LFP response to different simulation frequencies in the striatum and the FAF at depths of 300 μm **(g)** and 800 μm **(h)**. The red circles indicate the bf. **(i), (j), (k)** Mean LFP traces (±SEM) following auditory stimulation at the bf in the same exemplary recordings mentioned above. The arrows refer to the stimulation

onset. Subpanels **(l)**, **(m)**, **(n)** show the instantaneous energy of the respective LFP traces shown in (i)-(k). Data underlying this figure can be found at https://doi.org/10.12751/g-node.6a0d94. CN, caudate nucleus; FAF, frontal auditory field; LFP, local field potential.
(EPS)

**S4 Fig. Illustrative differences in LFP activity during vocalization in the FAF and CN. (a)** Mean LFP (±SEM) of all isolated communication calls (*n* = 31) during one recording in the striatum and **(b)** of all isolated echolocation pulses in the same recording (*n* = 28). **(c)** The mean ± SEM traces in the FAF at four representative depths (200, 400, 600, and 800 μm) in the same time period as (a) for communication and **(d)** echolocation. In the FAF, highest differences between call types occurred in the deepest channels. The example traces show activity before call onset, which could be used to predict the type of vocal output in both brain regions. Data underlying this figure can be found at https://doi.org/10.12751/g-node.6a0d94. CN, caudate nucleus; FAF, frontal auditory field; LFP, local field potential.
(EPS)

**S5 Fig. Example neural recordings showing different LFP frequency patterns in the FAF and CN.** In each subpanel, bottom panels show the broadcasted call, whereas top panels show filtered LFP traces obtained before and after call production in each case. **(a)** Individual example LFP in the CN during echolocation and **(b)** communication filtered in high gamma. **(c)** Example filtered LFPs (in the beta range) during echolocation and **(d)** communication. **(e)**–**(h)** Analogous exemplification for the FAF at 800-μm cortical depth. **(i)**–**(l)** Second example FAF recording during echolocation ((i) and (k)) and communication trials ((j) and (l)). In the FAF, during echolocation trials, high gamma and beta power occurs before call production. Data underlying this figure can be found at https://doi.org/10.12751/g-node.6a0d94. CN, caudate nucleus; FAF, frontal auditory field; LFP, local field potential.
(EPS)

**S6 Fig. Differences in LFP power during the production of LHF communication versus LF communication calls (see also S2 Fig). (a)** Bitmap of the Cliff's Delta effect size measure in the CN when comparing LHF communication calls (higher power in red) with LF calls (higher power in blue) revealing small power differences. **(b)** Similar as panel (a), but for the FAF at 300 μm and **(c)** 800-μm depth. Also in the FAF, the distinct pattern of power differences was less clear than when comparing communication calls to echolocation pulses (see Figs 5 and 6 of the main manuscript). **(d)** Average Cliff's Delta across FAF depths in the theta-alpha and **(e)** in gamma ranges. Here, the strongest size effect occurred in low alpha-theta (but not in gamma) in deep FAF channels at time points close to vocal production (0 in the x-axis). LFPs underlying this figure can be found at https://doi.org/10.12751/g-node.6a0d94. CN, caudate nucleus; FAF, frontal auditory field; LF, low-frequency; LFP, local field potential; LHF, low- and high-frequency.
(EPS)

**S7 Fig. Cross-validation and prediction accuracy of SVM prediction models. (a)** Cross-validation error across LFP frequencies in the CN and **(b)** in the FAF. Note that the lowest cross-validation errors occurred in the deep channels of the FAF in high gamma. **(c)** Assessment of the SVM model performance based on randomly chosen labels in the training sessions in the CN and **(d)** in the FAF. With unfaithful training information, model accuracy drops to values around a chance level, approximately 50%. Data underlying this figure can be found at https://doi.org/10.12751/g-node.6a0d94. CN, caudate nucleus; FAF, frontal auditory field; LFP, local field potential; SVM, support vector machine.
(EPS)

**S8 Fig. Coherence in fronto-striatal areas during the production of LF and LHF communication calls. (a)** Mean LFP coherence related to LHF communication calls between the CN and FAF at 300-μm depth and **(b)** 800-μm cortical depth. **(c)** Average coherence between the CN and different cortical depths of the FAF in theta and **(d)** alpha. Black lines indicate coherence values above the 95th percentile. For LHF calls, the highest coherence occurred before call onset in the low frequencies. **(e)–(f)** Same as (a)–(b) and **(g)–(h)** same as (c)–(d) but for LF communication calls. The coherence during LF communication call production was strongest in the same frequency band but shifted in time to early times (<250 ms) after call onset. Note that coherence in the echolocation condition occurred at later time points (>250 ms after call onset; see Fig 7 in the main manuscript). Data underlying this figure can be found at https://doi.org/10.12751/g-node.6a0d94. CN, caudate nucleus; FAF, frontal auditory field; LF, low-frequency; LHF, low- and high-frequency.
(EPS)

**S9 Fig. Illustrative calculation of phase locking values in one example trial within one brain structure. (a)** Spectrogram of the echolocation pulse emitted in this vocalization trial. **(b)** Top: spike times obtained in this trial in the CN (represented as dots). Bottom: simultaneously recorded raw LFP trace in the CN. Filtered LFP signals are shown in **(c)** theta, **(d)** alpha, **(e)** low-beta, **(f)** high-beta, **(g)** low-gamma, and **(h)** high-gamma. **(i)–(n)** Instantaneous phase values extracted after Hilbert-transforming the filtered LFP signals associated with panels (c)-(h), respectively. Phase values at the time points in which spiking occurred were used for phase-locking calculations. CN, caudate nucleus; LFP, local field potential.
(EPS)

**S10 Fig. Circular mean distributions illustrating spike-phase locking across brain regions and frequencies.** Circular mean distributions for communication call production in **(a)** theta, **(b)** alpha, **(c)** low beta, **(d)** high beta, **(e)** low gamma, and **(f)** high gamma. In each row, the first column indicates values acquired from the CN, whereas columns 2–4 display data obtained at three different FAF depths (i.e., 50 μm, 500 μm, and 800 μm). Red lines indicate the VS of each circular distribution. Data plotted in orange represent surrogate distributions. **(g)–(l)** Same as (a)-(h) but for the echolocation condition. dVS = difference in VS between the spike-phase locking distributions and the surrogate distributions; * $p < 0.001$ (Wilcoxon ranksum comparing VS values across randomization trials, Bonferroni corrected). Data underlying this figure can be found at https://doi.org/10.12751/g-node.6a0d94. CN, caudate nucleus; FAF, frontal auditory field; VS, vector strength.
(EPS)

**S11 Fig. Effect sizes obtained from comparisons of phase locking values. (a)** Cliff's Delta (*d*-values) obtained when comparing VS in the communication condition to the surrogate distributions in the CN across LFP frequencies bands ($n = 10,000$; l. = low; h. = high). **(b)** Same as (a), for the echolocation-surrogate condition. **(c)** *d*-Values obtained when comparing the VS of the echolocation and communication conditions (positive = higher phase locking before echolocation; negative = higher phase locking before communication). **(d)–(f)** *d*-Values across depths and LFP frequency bands in the FAF for the communication-surrogate condition, (e) echolocation-surrogate condition, and (f) echolocation-communication condition. *Indicates a small effect size (i.e., $d > 0.147$, found only in one case, see panel (e)). Data underlying this figure can be found at https://doi.org/10.12751/g-node.6a0d94.
CN, caudate nucleus; FAF, frontal auditory field; LFP, local field potential; VS, vector strength.
(EPS)

**S1 Table. Table depicting the main results of this study.** See also Fig 10 of the main manuscript. CN, caudate nucleus; FAF, frontal auditory field.
(EPS)

## Acknowledgments

We thank Gisa Prange for histological support, Christin Reißig for animal care, and Manfred Kössl for valuable comments on an earlier version of the article.

## Author Contributions

**Conceptualization:** Kristin Weineck, Francisco García-Rosales, Julio C. Hechavarría.

**Formal analysis:** Kristin Weineck.

**Funding acquisition:** Julio C. Hechavarría.

**Investigation:** Kristin Weineck, Julio C. Hechavarría.

**Methodology:** Kristin Weineck, Francisco García-Rosales, Julio C. Hechavarría.

**Project administration:** Julio C. Hechavarría.

**Resources:** Julio C. Hechavarría.

**Software:** Kristin Weineck, Francisco García-Rosales.

**Supervision:** Julio C. Hechavarría.

**Visualization:** Kristin Weineck.

**Writing – original draft:** Kristin Weineck.

**Writing – review & editing:** Francisco García-Rosales, Julio C. Hechavarría.

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
