## [Editor Report · Decision Letter 0]

11 Sep 2019

Dear Dr Hechavarria, 

Thank you for submitting your manuscript entitled "Neural oscillations in the fronto-striatal network predict vocal output in bats" for consideration as a Research Article by PLOS Biology.

Your manuscript has now been evaluated by the PLOS Biology editorial staff, as well as by an academic editor with relevant expertise, and I'm writing to let you know that we would like to send your submission out for external peer review.

Please re-submit your manuscript within two working days, i.e. by Sep 13 2019 11:59PM.

Kind regards,

Roli Roberts

Senior Editor

PLOS Biology

---

## [Decision Letter · Decision Letter 1]

31 Oct 2019

Dear Dr Hechavarria,

Thank you very much for submitting your manuscript "Neural oscillations in the fronto-striatal network predict vocal output in bats" for consideration as a Research Article at PLOS Biology. Your manuscript has been evaluated by the PLOS Biology editors, an Academic Editor with relevant expertise, and by three independent reviewers.

You’ll see that all three reviewers are largely positive about your study, but raise concerns about the framing (rev #1) and the nature of the distinction between different call types and their corresponding oscillations (revs #2 and #3). There are a substantial number of further requests for presentational improvement and additional analyses; all of the issues raised by the reviewers should be addressed for further consideration.

In light of the reviews (below), we will not be able to accept the current version of the manuscript, but we would welcome resubmission of a much-revised version that takes into account the reviewers' comments. We cannot make any decision about publication until we have seen the revised manuscript and your response to the reviewers' comments. Your revised manuscript is also likely to be sent for further evaluation by the reviewers.

Your revisions should address the specific points made by each reviewer. Please submit a file detailing your responses to the editorial requests and a point-by-point response to all of the reviewers' comments that indicates the changes you have made to the manuscript. In addition to a clean copy of the manuscript, please upload a 'track-changes' version of your manuscript that specifies the edits made. This should be uploaded as a "Related" file type. You should also cite any additional relevant literature that has been published since the original submission and mention any additional citations in your response. 

Before you revise your manuscript, please review the following PLOS policy and formatting requirements checklist PDF: http://journals.plos.org/plosbiology/s/file?id=9411/plos-biology-formatting-checklist.pdf. It is helpful if you format your revision according to our requirements - should your paper subsequently be accepted, this will save time at the acceptance stage.

Please note that as a condition of publication PLOS' data policy (http://journals.plos.org/plosbiology/s/data-availability) requires that you make available all data used to draw the conclusions arrived at in your manuscript. If you have not already done so, you must include any data used in your manuscript either in appropriate repositories, within the body of the manuscript, or as supporting information (N.B. this includes any numerical values that were used to generate graphs, histograms etc.). For an example see here: http://www.plosbiology.org/article/info%3Adoi%2F10.1371%2Fjournal.pbio.1001908#s5.

For manuscripts submitted on or after 1st July 2019, we require the original, uncropped and minimally adjusted images supporting all blot and gel results reported in an article's figures or Supporting Information files. We will require these files before a manuscript can be accepted so please prepare them now, if you have not already uploaded them. Please carefully read our guidelines for how to prepare and upload this data: https://journals.plos.org/plosbiology/s/figures#loc-blot-and-gel-reporting-requirements.

Upon resubmission, the editors will assess your revision and if the editors and Academic Editor feel that the revised manuscript remains appropriate for the journal, we will send the manuscript for re-review. We aim to consult the same Academic Editor and reviewers for revised manuscripts but may consult others if needed.

We expect to receive your revised manuscript within two months. Please email us (plosbiology@plos.org) to discuss this if you have any questions or concerns, or would like to request an extension. At this stage, your manuscript remains formally under active consideration at our journal; please notify us by email if you do not wish to submit a revision and instead wish to pursue publication elsewhere, so that we may end consideration of the manuscript at PLOS Biology.

When you are ready to submit a revised version of your manuscript, please go to https://www.editorialmanager.com/pbiology/ and log in as an Author. Click the link labelled 'Submissions Needing Revision' where you will find your submission record. 

Sincerely,

Roli Roberts

Senior Editor

PLOS Biology

REVIEWERS' COMMENTS:

Reviewer #1:

This manuscript describes a series of experiments in which local field potentials were simultaneously recorded in the striatum and frontal cortex of bats, and the patterns of neural activity were analyzed and compared before and after vocalizations were emitted. The manuscript focuses on comparing neural activity across two different types of vocalizations, echolocation calls versus communication calls. The LFPs are characterized and segregated by frequency into low (theta), medium (beta) and high (gamma) band activity. The study provides solid evidence of a functional interaction between neuronal activity patterns in frontal cortex and striatum, and found that activity patterns were different between the two types of vocalizations. Overall, the study is original, creative and the results are interesting. The presence of coherent premotor neural activity patterns in the frontal cortex and striatum preceding vocalizing is especially important since the role of the basal ganglia in non-human mammalian vocalizations remains poorly understood. The differences in the post-vocalization activity, which appears auditory in origin, are also quite intriguing.

My main criticism of the manuscript is that the hypothesis is not well justified, and the rationale for the study and its interpretation is vague and confusing. Most of the references cited in the introduction to build the rationale come from human speech and primate communication studies (3-14), with very little support for the presence of analogous brain networks in bats or other mammals (or songbirds). The stated hypothesis (Intro, 4th paragraph) is that the production of echolocation and non-echolocation (communication) calls could involve different fronto-striatal oscillatory dynamics. Predicting that the rhythms might be different during different behaviors isn’t a very satisfying test of the hypothesis unless the experiments is designed to reveal something specific about the underlying mechanism. The potential significance of different oscillatory frequencies or states for cognition and motor control is only generally addressed, and consequently the interpretation lacks any sense of mechanism. What seems to be absent is a cogent connection between corticostriatal oscillatory network activities and the mammalian (vocal) motor pathways. Without this, it is difficult to evaluate the significance of these results.

Specific comments

Intro page 2, line 5: what is the “hearing-action cycle”?

 Line 7: distress and social calls are communication calls. Why use the term non-echolocation rather than communication throughout the paper? 

Results, 2nd paragraph: For data analysis it makes sense to focus on utterances surrounded by a +/- 500 ms time window of silence, but wouldn’t this exclude normal echolocation behavior in which calls emitted in rapid trains? Don’t bats normally emit echolocation calls more frequently than once per second? 

Why did the bats emit communication calls if they were all alone in the recording chamber? Was there a method for evoking calls?

Results, 4th paragraph: The acoustic responses are puzzling. Line 5 says LFPs in both regions revealed a preference towards low frequencies (i.e those characteristic of the communication calls). If these neuronal populations weren’t very responsive to the echolocation call frequencies, why would they be involved in echolocation call generation? 

Page 8, first line: the observed effects would seem to support only a correlation between LFP frequencies and different vocalizations. A functional or causal role doesn’t seem to be supported yet.

Discussion, 2nd paragraph. Much of the discussion, but especially this paragraph (lines 9-19) seem highly speculative to me.

Page 14, line 19: “One possible explanation could be that echolocation calls require a more thorough sensory processing after vocal production than non-echolocation calls”. This would seem to go without saying, but it doesn’t really explain why there wasn’t an auditory response to the self-heard communication calls. Were there auditory responses to the acoustic stimuli, but no auditory responses during the communication call utterances? 

Page 16, line 7: The results don’t fully support the conclusion that neural activity patterns in FAF or CN can “represent future vocal outputs”, only that there is a correlation between network activity patterns and broad categories of vocal output, which may or may not be directly affiliated with motor control.

Reviewer #2:

General summary: Understanding the neural circuits underlying the production of vocal signals is an important endeavor, especially because humans are so reliant on speech and language for communication. In this study, Weineck et al. record neural activity in the frontal cortex and striatum in bats producing echolocation and non-echolocation calls. They find that neural activity and patterns of coherence in and between these brain areas varies between echolocation and non-echolocation calls. These are novel findings that have implications for the control of communication. 

Major comments:

It is important for readers to fully understand the extent of acoustic differences between echolocation (E) and non-echolocation (NE) calls included in their analyses. In this respect, it will be very important for the authors to depict, on the same figure, E and NE calls using both duration and peak frequency (2-D image). Based on the histograms, there is considerable overlap in both duration and peak frequencies, but these vocalizations could appear more distinct if each call is depicted using both features. One could also include frequency modulation as a third feature to help distinguish between E and NE calls.

Overall, the NE class of vocalizations is quite broad. The authors note in the Introduction that NE calls include communication, distress and social calls. It is unclear how acoustically distinct these types of NE calls are, or to what extent the NE calls in the analyses reflect these different types of NE calls. Given the differences between E and NE calls, it is possible that different types of NE calls are driven by different patterns of activity and coherence in the brain. Therefore, it seems important for the authors to parse their dataset of NE calls into different types of communication signals. One could also image it being interesting to assess the degree to which variation in signal properties within NE (and E) calls are related to LFP activity. At minimum, the authors should discuss the range of NE calls produced, the possibility of multiple types of NE calls emitted by the bats, and the limitations of their interpretations given the range of NE calls that could have been emitted during neurophysiological recordings. 

Related to this point, the authors should comment on the range of peak frequencies observed in E calls. The range seems quite large and it is important for a broad audience to understand the extent of this variation. 

One of the most consistent results in their paper is that many changes in neural activity and coherence are found after vocalizations. The authors explicitly acknowledge the prevalence of these results and suggest that these neural changes reflect processing of auditory feedback. While understanding the nature of auditory responses outside the context of vocalization is distinct from understanding the nature of auditory feedback processing, it would be very informative for the authors to conduct the same level of analyses of their data on auditory responses (e.g., LFP deflections, oscillations, coherence). This is useful even if the authors did not present a stimulus set that reflected the range of vocalizations emitted by bats. (If there are limitations on space, the authors could move much of the machine learning results to the Supplementary Information.)

The Introduction should provide more information on the potential importance of rhythms and coherence for action selection. I think much of the details about coherence and rhythms that is currently in the Discussion section should be moved to the Introduction. This will allow the reader to process the many types of data in the Results section.

The authors have plenty of additional data to analyze, since their (sensible) criteria for call selection dramatically winnowed down the number of calls for analysis. However, given their emphasis on elucidate neural activity that could lead to call production, it would be useful to independently analyze the data for echolocation and non-echolocation calls that were preceded by 500 ms of silence but did not have to be followed by 500 ms of silence. I think this would be a very useful dataset to test the robustness of their findings. 

The timing of the increase in LFP signals for echolocation calls is very late for auditory processing of a fast signal for detection objects in the world (Figure 2d). And the peak in coherence between the FAF and CN is even later, making it even less likely to be involved in auditory processing. The latency is also longer than that observed for auditory response to pure tones (Supplementary Figure 2). Please respond to this. 

Minor comments: 

One of the difficulties in trying to digest this paper is the large number of analyses (e.g., different frequency bands at different depths in the FAF with different vocalization types, coherence at different frequencies and depths across brain areas). It is difficult to keep everything in perspective, to know what is important. The Discussion and the graphical abstract do a pretty good job summarizing the data.

p. 7, first para, last sentence: since the increase in neural activity BEFORE the vocalization is not significant, this statement should be removed.

Section on SVM classifier is neat but think can be moved to supplementary information section for create space for additional analyses (see Major Comments). I also think the section on the relationship between spiking activity and the LFP is interesting, but see it as secondary to the main point of the paper; so this section too could be moved to the supplementary material (or streamlined substantially).

Figure 5. Why have two different depictions of accuracy across CN and FAF? Why not depict the data for CN as a heat map and add as a row onto the heatmap for FAF? Would save space and reduce the amount of work that readers have to do.

Please explain why opening and closing the recording chamber stimulated vocal production.

Reviewer #3:

[identifies himself as Jagmeet S. Kanwal]

This study reports interesting and novel findings on the role of oscillations for production and perception of vocalizations in bats. These results are significant given our lack of understanding of the neural mechanisms underlying vocal control. The role of oscillations in neural activity during social communication is an active area of inquiry and this study extends recent findings. While the oscillations are interesting, their correlation with vocal production does not inform us of the causal mechanisms and circuit design. Nevertheless, the findings reported here could pave the way forward. 

Technically, the study is well designed and executed on an important animal model in this field. 

Data appear to be carefully analyzed in detail and all figures are generally well organized, but the writing of the paper can be improved, including proper use of terminology. One main point that stands out is that social vocalizations predominantly have low frequencies, whereas echolocation vocalizations are stereotypically high frequency. Therefore, the differences in neural oscillations may be directly related to the dominant frequency in vocalizations and only secondarily to the fact that one set is being used for navigation while the other is used for social communication. For a clear distinction into two functional types, it is essential to compare social vocalizations that have high frequencies within them with echolocation vocalizations. Until this is done, the differences in neural oscillations correlated with each type of output cannot be classified as relating to social versus navigational (echolocation). The authors may be able to parse out their social vocalization data to clarify this issue.

Alternatively, the distinction between neural oscillations can be related to predominant frequency within a vocalization, which may very well be the case. 

Since different types of intra- and into inter-areal oscillations are discussed, to help the reader, please summarize results in a table where the function and/or presence/location of each type of oscillation and its relationship to communicaton/echolocation or high/low frequency vocalization is listed.

Also, it is important to be clear that the reader does not think that neural oscillations play a direct role in vocal control; the neural oscillations are temporally correlated with different types of vocal output. With regard to this, it will be useful to include a paragraph or section in the discussion about the origin of different types of oscillations in the brain in relation to this study (perhaps, expand on supplementary figure 9?). For example, high gamma oscillations are thought to result from the interaction of fast-spiking inhibitory with excitatory neurons within local neural circuits.

Also, oscillations observed in one region of the brain can originate elsewhere since they reflect synaptic activity. This can have a bearing on the proposed fronto-striatal connectivity. 

Specific corrections:

Abstract:

First line: Change "… networks leading to vocalization production remain…" to "… networks underlying vocal control remain…"

Change "... allows to assess the ... " to " ... allowed us to assess general... "

Improve ".... the precise control of high gamma…" This is not good expression as the high gamma is not being controlled, but the vocal production is being controlled.

Change ".. Our data indicate.., " ("data" is plural). 

Change "... animals to access ... " to "... animals to selectively activate motor programs required for the production of either echolocation or social vocalizations. "

Change to: "… animals are in a navigational mode (i.e., emitting echolocation pulses) or in a social communication mode (emitting social calls). Overall, this study presents…"

Introduction:

Change to "... motor output has not been clearly delineated." Also please report here the extensive work of Jurgens on neural pathways for vocal control in primates.

Clarify " a largely discussed how…". Something is missing here.

2nd para: here and elsewhere, delete "production". "Vocalization" implies production.

Clarify "associative circuit" - does not seem to be directly relevant here.

"... latter could suggest…" – Delete "could"

P. 4: Here and elsewhere: Avoid use of the term "echolocation calls" and "non-echolocation calls" 

Also, "Non-echolocation calls such as communication, distress and social calls". This is confusing. Better to have clear separation and consistently use "echolocation pulses or vocalizations". Non-echolocation vocalizations should instead clearly refer to "social calls". Distress calls also fall under the category of social calls. These are the only two well-recognized types of vocalizations. Non-echolocation calls is a functionally diffuse category. 

P. 7: Please briefly indicate the importance of Cliff's delta metric and why it was used here to compare effect size.

Change to: "… before call production compared to before emission of echolocation pulses…" 

Last sentence on page 11 is not clear.

Not clear why the phase of LFPs in the FAF is compared to its relationship to striatal spiking. Please explain. Also, it is not clear why changes in spiking probability were not observed in response to vocalizations. Could this be because spiking activity was not recorded from relevant neurons given the sparse nature of their distribution within the FAF?

Discussion:

As used elsewhere, change "diseases" to "disorders".

Last sentence on page 12: please state more clearly. Which motor action? Also, visual abstract in supplemental figure 9 can be improved in terms of labels and colors and clarity in general.

P. 14: please provide some reasoning for the first sentence in the second paragraph on this page.

P. 16: change "component" to "outcome". 

P. 28, Figure 2 legend: "z-scored LFPs". Does this refer to the peak of each LFP?

P. 31: "... for different channels…". Should this be "... for all recorded channels..."?

---

## [Decision Letter · Decision Letter 2]

20 Jan 2020

Dear Dr Hechavarria,

Thank you for submitting your revised Research Article entitled "Neural oscillations in the fronto-striatal network predict vocal output in bats" for publication in PLOS Biology. I have now obtained advice from two of the original reviewers and have discussed their comments with the Academic Editor. 

We're delighted to let you know that we're now editorially satisfied with your manuscript. However before we can formally accept your paper and consider it "in press", we also need to ensure that your article conforms to our guidelines. A member of our team will be in touch shortly with a set of requests. As we can't proceed until these requirements are met, your swift response will help prevent delays to publication. Please also make sure to address the data and other policy-related requests noted at the end of this email.

*Copyediting*

*Published Peer Review History*

*Early Version*

*Submitting Your Revision*

Sincerely,

Roli Roberts

Senior Editor

PLOS Biology

ETHICS STATEMENT:

The Ethics Statements in the submission form and Methods section of your manuscript should match verbatim. Please ensure that any changes are made to both versions.

-- Please include the full name of the IACUC/ethics committee that reviewed and approved the animal care and use protocol/permit/project license. Please also include an approval number.

-- Please include the specific national or international regulations/guidelines to which your animal care and use protocol adhered. Please note that institutional or accreditation organization guidelines (such as AAALAC) do not meet this requirement.

-- Please include information about the form of consent (written/oral) given for research involving human participants. All research involving human participants must have been approved by the authors' Institutional Review Board (IRB) or an equivalent committee, and all clinical investigation must have been conducted according to the principles expressed in the Declaration of Helsinki.

DATA POLICY:

Regardless of the method selected, please ensure that you provide the individual numerical values that underlie the summary data displayed in the following figure panels as they are essential for readers to assess your analysis and to reproduce it: Figs 1-9, S2-S8, S10, S11. NOTE: the numerical data provided should include all replicates AND the way in which the plotted mean and errors were derived (it should not present only the mean/average values).

REVIEWERS' COMMENTS:

Reviewer #1:

I like the changes to the introduction and discussion. Even though I'm still not sure how these oscillations relate to motor output or decision making, I'm convinced that the manuscript has addressed this issue to the best that it can be handled with current knowledge. The revision has a clearer picture of the hypothesis and goals, and the discussion is more coherent and understandable. I am satisfied that the revisions have addressed my earlier concerns and I have no further comments. 

Reviewer #3:

[identifies himself as Jagmeet S. Kanwal]

The authors have done an outstanding job of addressing reviewer comments. They conducted additional analyses, created and improved figures and a table as suggested, and rewrote parts of the manuscript and improved terminology. The manuscript reads quite well now and reports some interesting findings and stimulating ideas for further work.

---

## [Editor Report · Decision Letter 3]

13 Feb 2020

Dear Dr. Hechavarria,

On behalf of my colleagues and the Academic Editor, Dr. Manuel S. Malmierca, I am pleased to inform you that we will be delighted to publish your Research Article in PLOS Biology. 

Early Version

PRESS 

Kind regards,

Krystal Farmer

Development Editor, 

PLOS Biology

on behalf of

Roland Roberts,

Senior Editor

PLOS Biology